# Pushing the Limits of All-Atom Geometric Graph Neural Networks: Pre-Training, Scaling and Zero-Shot Transfer

**Zihan Pengmei** [*]
The University of Chicago
zpengmei@uchicago.edu

**Zhengyuan Shen**
Amazon Web Services
donshen@amazon.com

**Zichen Wang**
Amazon Web Services
zichewan@amazon.com

**Marcus Collins**
Amazon
collmr@amazon.com

**Huzefa Rangwala**
Amazon Web Services
rhuzefa@amazon.com

## Abstract

The ability to construct transferable descriptors for molecular and biological systems has broad applications in drug discovery, molecular dynamics, and protein analysis. Geometric graph neural networks (Geom-GNNs) utilizing all-atom information have revolutionized atomistic simulations by enabling the prediction of interatomic potentials and molecular properties. Despite these advances, the application of all-atom Geom-GNNs in protein modeling remains limited due to computational constraints. In this work, we first demonstrate the potential of pre-trained Geom-GNNs as zero-shot transfer learners, effectively modeling protein systems with all-atom granularity. Through extensive experimentation to evaluate their expressive power, we characterize the scaling behaviors of Geom-GNNs across self-supervised, supervised, and unsupervised setups. Interestingly, we find that Geom-GNNs deviate from conventional power-law scaling observed in other domains, with no predictable scaling principles for molecular representation learning. Furthermore, we show how pre-trained graph embeddings can be directly used for analysis and synergize with other architectures to enhance expressive power for protein modeling.

## 1 Introduction

*In silico* molecular computation and simulation are indispensable tool in modern research for biology, chemistry, and material sciences capable of accelerating the discovery of new drugs and materials, as well as prediction of protein structures and functions. One of the key advance in molecular computation is the use of learnable geometrical descriptors for representing the atomic environment (Todeschini & Consonni, 2008; Mauri et al., 2017; Baig et al., 2018; Moriwaki et al., 2018). Geometric Graph Neural Networks (Geom-GNNs) with all-atom resolutions can learn to construct system-specific molecular force fields by incorporating geometric inductive biases such as rotational invariance (Schütt et al., 2018; Gasteiger et al., 2020) and equivariance (Schütt et al., 2021; Batzner et al., 2022). These ML-based force fields can be seen as learnable geometric descriptors that map the molecular conformation to the potential function of a target. These capabilities position Geom-GNNs as powerful tools for mapping molecular conformations to target functions, such as potential energy surfaces. However, most applications remain restricted to quantum chemistry tasks, with limited exploration in protein modeling. And such all-atom information can be crucial in modeling protein-protein interaction (PPI), protein-ligand binding, and protein reaction kinetics.

In contrast to other domains such as text and vision, where pre-trained models have been recognized as highly effective feature extractors trained with self-supervised objectives (Devlin et al., 2019; Radford et al., 2021; Kirillov et al., 2023), the potential of pre-trained Geom-GNNs for protein modeling remains underexplored. Recent advances, such as denoising pre-training, have shown promise in enhancing the generalization of Geom-

---

[*] Work done during an internship at Amazon Web Service.

GNNs across diverse molecular systems (Zaidi et al., 2022). Building on these developments, we first demonstrate that pre-trained Geom-GNNs can serve as effective zero-shot learners for protein systems, extracting rich, transferable representations when pretrained on small-molecule datasets, thereby addressing the challenges discussed earlier. We evaluate the performance of these pre-trained models by analyzing neural scaling laws (Kaplan et al., 2020; Hoffmann et al., 2022). Specifically, we studied pre-training graph embeddings *via* characterizing the graph neural scaling laws in the self-supervised learning pre-training, supervised regression and unsupervised feature extraction setups.

Our results reveal that, contrary to the power-law scaling observed in other domains, Geom-GNNs exhibit unique scaling patterns characterized by early saturation and diminishing returns. Furthermore, we explore how pre-trained all-atom graph embeddings can be integrated with other neural architectures to improve their expressive power in complex tasks such as protein folding and kinetic modeling. We additionally show how those pre-trained graph embeddings could be easily combined with other architectures to enhance their expressive power for protein modeling. The low-dimensional projection of the pre-trained graph embedding space (Figure 2 and Appendix K) shows meaningful clustering of similar conformations. These findings not only consolidate our understanding of Geom-GNN scaling principles but also pave the way for their broader application in molecular and protein sciences. Our contributions can be summarized as follows:

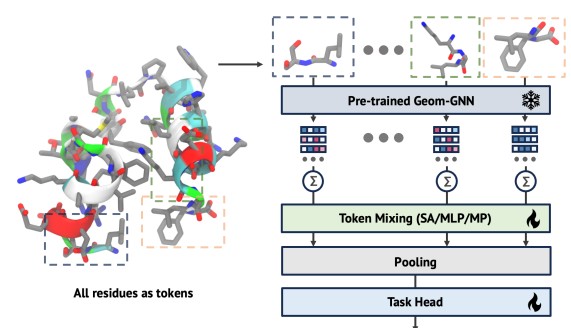

Figure 1: **Meta-architecture for Using Pre-trained Geom-GNNs as descriptors**: The figure shows a framework where pre-trained Geom-GNNs act as local geometric descriptors to featurize residue-level conformations. Each window represents an atomic environment for residue feature extraction, defined by a user-defined context (illustrated here as a sliding window of nearest neighbors in sequence). In each window, atomic structures are treated as individual graphs and processed by the pre-trained Geom-GNN to extract atomic-level features, which are aggregated into residue-level representations or "tokens." The architecture can employ self-attention (SA), multi-layer perceptron (MLP), or message passing mechanisms to enhance representational power. For graph-level tasks, the mixed tokens are pooled and input to a task-specific head for training and predictions.

We first demonstrate the use of pre-trained Geom-GNNs as zero-shot transfer learners for protein featurization when pre-trained on small-molecule datasets. When representing residue conformations in proteins with nearest neighbors, graph embeddings match conventional rotamer features but possess better expressivity in describing the atomic environment flexibly. The pre-trained graph embeddings excel at constructing kinetic models through the VAMPNet objective: a challenging unsupervised learning task that is crucial for enhanced sampling and probing energy landscapes in computational chemistry and biology. Compared to traditional pairwise distance features, the graph embeddings achieve significant performance gains up to 50% across complex peptide and protein systems.

We systematically investigated the scaling behaviors of Geom-GNNs under the self-supervised, supervised and unsupervised setups. We found that Geom-GNNs do not exhibit typical power-law scaling during self-supervised pre-training, suggesting distinct scaling principles for molecular representations compared to areas like language modeling. Under more rigorous setups requiring out-of-distribution (OOD) generalization, we found there is no predictable scaling behavior on molecular property and force field prediction tasks, in contrast to earlier discoveries. In the transfer learning setups for protein modeling, we also found there is no predictable scaling behavior in kinetic model construction and fold classification.

## 2 RELATED WORKS

**Pretraining *via* Denoising** is a novel pretraining strategy for equivariant GNN architectures, drawing inspiration from advancements in visual generative models (Sohl-Dickstein et al., 2015; Song et al., 2020a;b). Zaidi et al. (2022) introduced a method of corrupting input atomic coordinates of equilibrium molecular structures

and training the equivariant GNN to denoise and restore the original coordinates. The denoising objective is defined as: $L_{\text{Denoising}} = \mathbb{E}_r \|\epsilon - \text{GNN}(\hat{r})\|^2$. Here, $\epsilon$ represents the noise added to the original coordinates $r$, producing the noised input $\hat{r}$. Wang et al. (2023) expanded this method to include denoising of non-equilibrium structures. Further studies have investigated varying noise distributions such as sliced noise (Ni et al., 2023).

**Graph neural scaling laws** represent efforts to extend scaling laws from language modeling to the graph domain. Frey et al. (2023) demonstrated the power-law relationship between supervised loss, model size $N$, and dataset size $D$ when training a Geom-GNN as a potential function. Chen et al. (2024) further advanced this area by demonstrating a predictable power-law relationship between supervised loss and dataset size using a topological GNN. More recently, Liu et al. (2024) expanded the discussion to a parametric perspective, focusing on excessively large datasets such as PCQM4M (Nakata & Shimazaki, 2017; Hu et al., 2020) in a supervised setting without pre-training.

**Learning-based molecular dynamics** In molecular simulations, addressing complex tasks beyond basic molecular property predictions involves techniques such as kinetic modeling using time-lagged independent component analysis (TICA) (Molgedey & Schuster, 1994; Naritomi & Fuchigami, 2011; Scherer et al., 2015) and Variational Approach for Markov Process (VAMP) (Wu & Noé, 2020; Mardt et al., 2018), prediction of committer functions Strahan et al. (2023a;b) in transition path theory (Weigend, 2006; Metzner et al., 2009), and the application of enhanced sampling methods (Bonati et al., 2021; Jung et al., 2023). These tasks require meticulously crafted descriptors that capture the structural nuances of the systems under study. Recent advancements have underscored the value of atomistic detail, leading to the integration of learnable Geom-GNNs (Klein et al., 2024; Huang et al., 2024; Liu et al., 2023) into aforementioned more advanced modeling scenarios. However, the scalability of these methods remains challenging due to the necessity of training graph representations on-the-fly.

## 3 EXPERIMENTAL AND DATASET SETUP

We focus on two state-of-the-art architectures based on message-passing (MP): Equivariant Transformer (ET) (Thölke & De Fabritiis, 2022) and its extension, ViSNet (Wang et al., 2024), which incorporates higher body-order features. To investigate scaling effects, we separately studied effect of width, depth and aspect ratio of MP layers. We further analyze the effect of radius cutoff distance and refer readers to Appendix F for detailed model configurations and Table D for a summary of datasets used.

**Pre-training setup.** For pre-training, we employed a vanilla denoising objective, where scaled noise was added directly to the input atomic coordinates, and the network was tasked with predicting this added noise. After pre-training, the denoising head was discarded. We utilized two distinct datasets for pre-training: PCQM4Mv2 (Nakata & Shimazaki, 2017; Hu et al., 2020) and the training dataset of OrbNet-Denali (Christensen et al., 2021) (hereafter referred to as PCQM and Denali). These datasets represent equilibrium and non-equilibrium molecular structures, respectively. In PCQM, each sample represents a unique molecular topology, with structures relaxed using DFT, thus providing equilibrium structures. Conversely, Denali contains multiple geometric conformations for each molecular topology, offering a diverse range of non-equilibrium structures.

**Supervised downstream tasks setup.** The supervised downstream tasks to validate the learned representations encompass diverse molecular structures and conformations. For chemical structure diversity, we utilized the QM9 dataset (Ramakrishnan et al., 2014) for molecular property prediction, implementing an ID setup through random splitting (Random-QM9) and an OOD setup through scaffold splitting based on molecular motifs using RDKit (Landrum et al., 2013) (Scaffold-QM9). For conformational diversity, the xxMD dataset (Pengmei et al., 2024a) was employed, which surpasses the MD17 benchmark (Chmiela et al., 2017) in sampling and computational techniques. This dataset features extensively sampled non-equilibrium conformations within reactive regions. We applied a temporal split (Temporal-xxMD) based on timesteps and a random split (Random-xxMD) as ID and OOD scenarios. Here, Random-xxMD represents a well-sampled condition, ensuring even representation of conformational space across training and testing sets. Conversely, Temporal-xxMD presents a more challenging scenario with less time correlation and poorer sampling, demanding significant extrapolation capabilities.

**Zero-shot transfer setup.** Beyond conventional supervised learning, we explored zero-shot transfer approaches to extract pre-trained graph representations, following the methodology outlined in Mardt et al. (2018) and Pengmei et al. (2024b). For kinetic modeling tasks for characterizing non-reversible dynamics, we utilized

molecular dynamics trajectories from three systems: alanine dipeptide (ala2), pentapeptides from MDshare (Nüske et al., 2017; Wehmeyer & Noé, 2018), and λ6-85 fast-folding protein (Bowman et al., 2011). We inferred and stored activations from the last MP layer of pre-trained all-atom Geom-GNN checkpoints. These features are used to optimize the network as shown in Figure 1 with the VAMP score. To further assess the versatility of pre-trained representations, we investigated their potential to enhance other GNN architectures. Specifically, we augmented a previously developed protein-specific GNN (Wang et al., 2022) with activations from the pre-trained all-atom Geom-GNN. We then evaluated this enhanced model on a protein fold classification task (Hou et al., 2018).

## 4 ZERO-SHOT TRANSFER DOWNSTREAM TASKS

Comparing supervised objectives, self-supervised objectives are emerging areas of researches in learning-based molecular dynamics simulations. As mentioned before, training all-atom graph representations on-the-fly are subject to scalability and generalization issues. We propose to address these challenges by pre-trained graph embeddings including atom-level information, thereby separating the training of Geom-GNNs from the simulation objectives. This section is structured as follows: We begin by showcasing the application of zero-shot transferable Geom-GNN representations, integrated with other architectures to meet the VAMP objectives (Mardt et al., 2018), which aims to learn a low-dimensional representation that preserves the long-timescale kinetics of the molecular system. It is crucial in computational biology for enhanced sampling and understanding the underlying energy landscapes. We explored three systems with increased complexity up to 1258 heavy atoms: ala2, pentapeptide, and the λ6-85. Subsequently, we demonstrate how these features can be combined with protein-specific GNNs to improve the classification performance on protein folding, with analysis presented in Appendix K for brevity.

### 4.1 KINETIC MODELING: VAMP

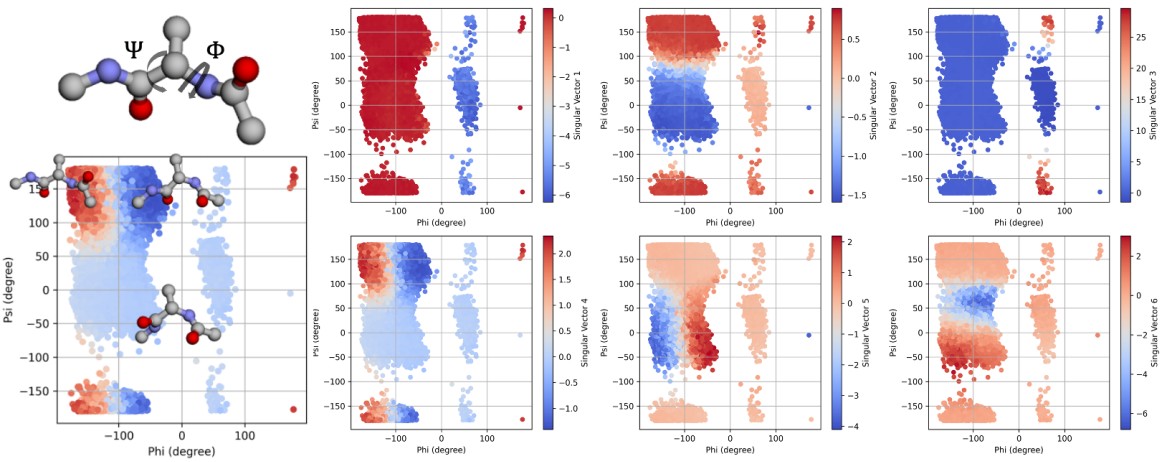

Figure 2: Visualization of learned singular vectors of Koopman operator onto the $\Psi - \Phi$ dihedral angle space of ala2 using pre-trained ViSNet embedding with 64 width. Those singular vectors describe the slow modes of the underlying dynamics (Appendix J).

Initial experiments focused on the ala2 system, known for its metastable states characterized by two main backbone dihedrals, $\Psi$ and $\Phi$. We input the atomic numbers and Cartesian coordinates of ten heavy atoms into the pre-trained ViSNet and extracted graph-level embedding. A separate VAMP head was then trained to learn the kinetic model. The learned coordinates were projected onto a two-dimensional space constructed by $\Psi - \Phi$ dihedrals. As illustrated in Figure 2, the network successfully identified the first learned singular vector as the slowest transition between positive and negative $\Psi$ angles. The subsequent vector characterized the transition between small and large $\Phi$ angles. Lower-order singular vectors identified transition states between the $\alpha$ and $\beta$

regions of the Ramachandran plot. These observations show that the graph representations capture structurally accurate and semantically meaningful information as illustrated in Figure 2, 14 and 17.

Koopman matrix quantifies the kinetic variance between instantaneous and time-lagged data (Appendix J). A minimal VAMP score of 1 corresponds to equilibrium, while the maximum score is theoretically determined by the dimensionality of the Koopman operator constructed, with the highest possible score being the output dimension plus one. However, the actual VAMP score varies depending on the characteristics of the molecular trajectories and the selected lag time. Figure 3 shows the VAMP scores for models trained on pentapeptide trajectories across various output dimensions and lag times, following the PyEMMA setup Scherer et al. (2015). Furthermore, to interpret the learned coordinates, we visualized characteristic structures and their projections onto conventional physical collective variables in Figures 14, 15 and 16 (Appendix). These visualizations facilitate direct comparisons with established reference literature (Scherer et al., 2015; Bowman et al., 2011). Our observations reveal that increasing the output dimension to 10, across all five tested lag times, does not plateau in performance when compared to the linear TICA method, which utilizes backbone torsion angles. This demonstrates that using graph features allow for the higher resolution comparing to conventional descriptors.[*]

Table 1: Validation of VAMP-2 scores (↑) for Ala2, Pentapeptide, and $\lambda$6-85 with PCQM-ViSNet with varying dimensions and model architectures, with a constant batch size of 5000. Scores reflect the performance with and without ('Sum' refers to non-learnable sum. pooling) residue-level token mixing in graph embedding vectors post-coarse-graining. Lag times: Ala2 at 1 ps, Pentapeptide at 0.5 ns, and $\lambda$6-85 at 25 ns. Except Ala2, all systems are split by trajectories, where half of trajectories are reserved for validation. 'OOM' refers to out-of-memory at this combination of dimension and batch size on an instance with $4 \times$ NVIDIA A10G GPUs. We reproduced Mardt et al. (2018) with the same setup, for ala2 and pentapeptide we consistently used the heavy atom pairwise distances, and we used pairwise distances of 80 $C_\alpha$ atoms for $\lambda$6-85.

| Dimension | Ala2 | Pentapeptide | | | $\lambda$6-85 | | |
|---|---|---|---|---|---|---|---|
| Token Mixing | Sum | Sum | MLP | SA | Sum | MLP | SA |
| **Out. Dim.** | **6** | **10** | | | **10** | | |
| **#Heavy Atoms** | **10** | **44 (5 residues)** | | | **1258 (80 residues)** | | |
| 64 | 4.71±0.01 | 5.63±0.37 | 7.00±0.47 | 4.81±1.52 | 4.27±0.06 | 8.49±0.24 | 7.01±0.21 |
| 128 | **4.72±0.01** | 6.41±0.07 | 6.91±0.32 | 7.11±0.43 | 5.17±0.08 | 8.35±0.36 | 7.80±0.18 |
| 256 | 4.70±0.01 | 6.10±0.22 | 7.67±0.08 | 7.04±0.21 | 4.75±0.26 | **8.52±0.14** | OOM |
| 384 | 4.70±0.01 | 6.62±0.16 | **7.71±0.23** | 7.26±0.11 | 5.81±0.18 | OOM | OOM |
| Mardt et al. | 3.92±0.47 | 5.14±0.25 | | | 7.40±0.40 | | |

In terms of scaling feature dimensions, We observed no significant performance gains for Ala2 when increasing the feature dimension beyond 64, as the model effectively captured all critical slow dynamics at this lower dimension. However, for more complex systems like Pentapeptide and $\lambda$6-85, which possess a greater number of degrees of freedom, scaling the embedding dimension provided clear performance improvements. This indicates that the effectiveness of scaling is contingent upon the complexity of the system and whether the network has reached its representational limit. Other than scaling, another key factor affecting performance is the use of token mixing techniques, where residue-level embeddings were generated from pre-trained all-atom graph embeddings and subsequently mixed using higher-level operators such as the MLP-Mixer and self-attention mechanisms (Pengmei et al., 2024b). As shown in Table 1, this approach led to significant performance improvements, particularly for $\lambda$6-85, where applying the MLP-Mixer resulted in over a 46% increase in VAMP-2 scores compared to the baseline sum pooling method.

In comparison to the baseline VAMPnet model (Mardt et al., 2018), our approach using pre-trained graph features led to performance improvements of 20.4%, 50.0%, and 15.1% for Ala2, Pentapeptide, and $\lambda$6-85, respectively. In smaller systems such as the Pentapeptide, early-stage information compression can occur, and token mixing methods alleviate this limitation. For larger systems like $\lambda$6-85, the improvements gained from pre-trained features were not as substantial as for smaller systems. This indicates that future work should focus on developing more expressive models capable of generalizing global relationships from local tokens, as well

---

[*]`http://www.emma-project.org/latest/tutorials/notebooks/00-pentapeptide-showcase.html`

as refining the process of localizing graph embeddings for each residue in protein systems along molecular trajectories. Current methods, which treat the entire protein as a single graph, may suffer from over-smoothing, leading to the mixing of global and local features at each node. By addressing these challenges, more effective residue-level token mixing and introducing global features could be achieved to improve overall performance. In principle, this approach can be extended to study phase diagrams and free energy surfaces of polymeric and other complex chemical systems but we limit our scope to biomolecules due to the lack of reference.

## 4.2 PROTEIN REPRESENTATION LEARNING: FOLD CLASSIFICATION

As we explored the usage of pre-trained graph embedding to study the conformation space of peptides and proteins, we extend to explore the utility of such all-atom embedding in the protein representational learning. Depicted in Figure 1, modeling the dependence among multiple inputs require an active token mixer. MP-based GNNs themselves are also a type of token mixers. In protein-specific GNNs, residues are represented as nodes, and the features attached to each node are usually $C_\alpha$ positions and a set of geometrical descriptors for backbone or side-chain dihedral and torsion angles ("rotamers"). While these features are effective for describing secondary structures, they are inadequate for describing protein tertiary structures, meaning the lack of direct modeling of interactions between side-chains, particularly in environments with $\pi$-stacking between aromatic residues. More importantly, torsion angles do not capture hydrogen bonding between distant residues, salt bridges or disulfide bonds, which are vital for the tertiary structure identification. The direct modeling of the possible side-chain with solvent, membrane and ligand interaction is also missing, which can be addressed by Geom-GNN representation. For illustration, we naively concatenate projections

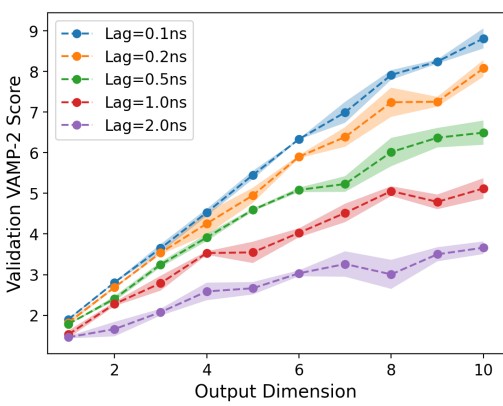

Figure 3: Validation VAMP-2 scores using pre-trained ViSNet embeddings with 256 length (no token mixer) across various output dimensions and lag times for the pentapeptide system. Random half of available trajectories are held for validation and the remaining are used for training.

of pre-inferred all-atom embedding to interaction blocks of Wang et al. (2022) and test the downstream task performance on the FOLD dataset (Hou et al., 2018). Per Figure 20, we observe improvement of integrating such all-atom features into the ProNet without angle information. More importantly, the pre-trained graph embedding can match with conventional rotamer features in non-interacting cases as shown in Figure 17, 18 and 19, where we used a sliding window on the backbone sequence. And it remains future efforts to curate such benchmarks that test how side-chains interact with environments.

## 5 SELF-SUPERVISED PRE-TRAINING TASKS

As we have studied a few preliminary application of pre-trained graph embedding, we wonder if the power of features given by Geom-GNNs with varying architectures and configurations. We approach this by first varying the network width with constant depth and cutoff radius, followed by separate assessments of depth and cutoff radius impacts. We also analyzed the effects of aspect ratio while maintaining a consistent total parameter count considering the specific challenges of MP-based architectures, including under-reaching and over-smoothing.

**Effect of model width.** We investigated the scaling behavior of ET and ViSNet on both PCQM and Denali datasets. For the denoising pre-training task, we measured performance using the mean square error (MSE) between the predicted and normalized noise. We varied model width and allowing each model to train 10 epochs. We report both the first epoch loss (where each data point is seen once) and the converged loss as parameter-limited and data-limited scenarios. Figure 6 illustrates findings, revealing: (1) Deviation from power-law: The scaling relationship between pre-training loss and model size does not strictly follow a power-law. (2) Rapid initial improvement: In low parameter regimes, model performance improves quickly as size increases. (3) Saturation effect: Beyond a certain threshold (e.g., after $10^7$ parameters), performance gains

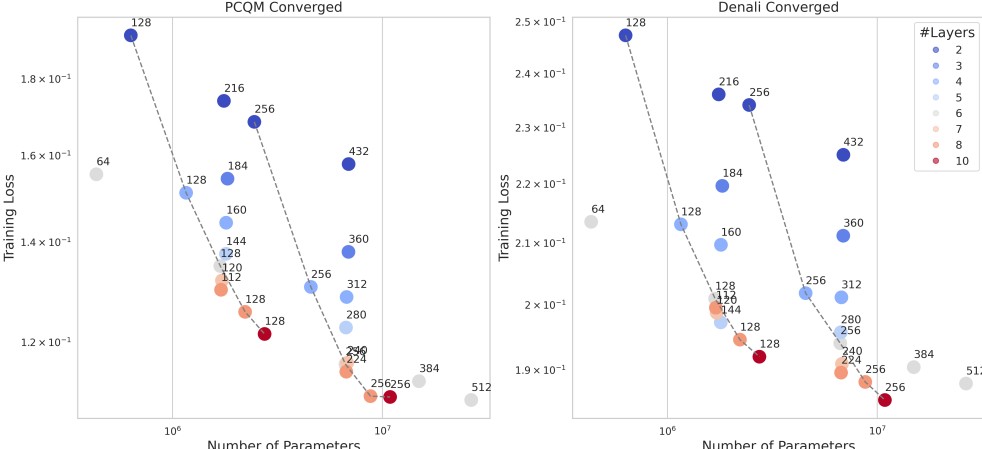

Figure 4: Scaling behavior of ViSNet depth for the converged results of training on PCQM and Denali datasets. The plot shows the relationship between the number of layers and the pre-training loss, illustrating the initial rapid improvement followed by diminishing returns as depth increases. We additionally show the results of the first epoch loss in Figure 7 (Appendix).

become marginal, with models often reaching a point where further increases in model size yield diminishing returns, deviating from the fitted power-law line.

**Effect of Model Depth.** The number of MP layers determines the receptive field size, as each layer gathers information from adjacent nodes, enabling data propagation from distant nodes. In experiments, we varied the network's width and depth to assess their impacts on pre-training loss, as depicted in Figures 4 and 7. Our results show significant performance gains with increased depth in the initial layers, with diminishing returns beyond 6 layers. Notably, scaling behaviors differ between the PCQM and Denali datasets. The PCQM dataset shows saturation in both initial and converged losses, whereas in the Denali dataset, deeper and narrower models tend to surpass wider and shallower ones, likely due to the larger average graph sizes (PCQM: 29.5 nodes, Denali: 45.0 nodes). Larger graphs benefit from more MP steps, but the over-smoothing issue inherent in MP-based GNNs limits the advantages of greater depth. With deeper networks, the influence of depth diminishes, and the model's performance becomes more reliant on increasing parameter counts. This shift suggests that optimal depth achieves maximum performance before capacity expansion becomes more effective.

**Effect of aspect ratio.** To further discern the effect of scaling the model depth, we fix the total parameter count by varying the model width to control the aspect ratio. By eliminating the parameterization effect, we observe the checkpoints with similar amount of parameters and lower aspect ratio (width/depth) enjoy better pre-training performance per depicted in Figure 8 (Appendix). In other words, deeper models are favored, though such improvement diminished as the depth is saturated.

**Effect of Cutoff Radius.** The cutoff radius in Geom-GNNs critically defines the model's effective receptive field by regulating the scope of atomic interactions during message passing. Figure 9 (Appendix) shows significant performance fluctuations at different radii. On the PCQM and Denali datasets, the pre-training loss for a 6-layer ViSNet model displays a U-shaped curve, decreasing to a minimum at approximately 5 Å before rising again up to 9 Å. This pattern indicates an optimal cutoff radius that effectively integrates local and global information, with more pronounced effects in the larger, more complex Denali dataset. In this dataset, losses increase sharply beyond the optimal radius, indicating the importance of distinguishing relevant local interactions from misleading long-range correlations.

## 6  SUPERVISED DOWNSTREAM TASKS

As described in the experimental setup, we additionally used two molecular benchmarks to verify the performance of learned features: QM9 for molecular properties prediction (chemical diversity) and xxMD for force

field prediction (conformational diversity). Our analysis covers two scenarios: transferring weights from pre-trained model and fine-tuning all parameters, and training models with identical configurations from scratch.

Table 2: Performance of ViSNet and ET models on the Scaffold-QM9 dataset with varying model widths. The table shows the mean absolute error (MAE) vs model width, comparing pre-trained (fine-tuned) models using PCQM dataset with those trained from scratch. All models are trained without the auxiliary denoising loss.

| Model | Dim | $\mu$ (mD) | $\alpha$ ($ma_0^3$) | $\epsilon_{\mathbf{HOMO}}$ (meV) | $\epsilon_{\mathbf{LUMO}}$ (meV) | $\Delta\epsilon$ (meV) |
|---|---|---|---|---|---|---|
| **PT-ViSNet** | **64** | 28.8 | 99.1 | 56.8 | 37.2 | 90.2 |
| | **128** | 24.9 | 93.4 | 56.8 | 35.9 | 80.7 |
| | **256** | 25.1 | 110.8 | 51.7 | **32.3** | 77.8 |
| | **384** | **23.2** | 91.2 | **47.1** | 34.8 | 77.0 |
| | **512** | 26.5 | **90.5** | 54.9 | 35.5 | **71.0** |
| **ViSNet** | **64** | 36.5 | 142.1 | 74.4 | 46.6 | 110.5 |
| | **128** | **32.3** | 106.4 | 62.0 | 42.8 | 94.5 |
| | **256** | 36.3 | **101.5** | **55.0** | **41.8** | **92.5** |
| | **384** | 34.3 | 115.2 | 60.9 | 44.2 | 96.2 |
| | **512** | 35.9 | 139.9 | 60.0 | 52.5 | 123.6 |
| **PT-ET** | **64** | 36.9 | 108.3 | 69.0 | 45.8 | 96.3 |
| | **128** | 33.5 | 117.4 | 62.1 | 40.2 | 88.4 |
| | **256** | **29.2** | 116.4 | 54.0 | **36.4** | 84.0 |
| | **384** | 35.2 | **101.3** | **52.3** | 40.1 | 82.0 |
| | **512** | 29.4 | 197.8 | 57.7 | 37.5 | **80.1** |
| **ET** | **64** | 35.9 | 104.2 | 72.7 | 47.5 | 101.9 |
| | **128** | 32.3 | **98.4** | 65.4 | 44.6 | **96.2** |
| | **256** | **31.3** | 127.5 | 65.1 | **43.2** | 100.9 |
| | **384** | 34.8 | 116.6 | **60.6** | 44.0 | 98.7 |
| | **512** | 53.8 | 178.7 | 83.7 | 53.3 | 97.7 |

## 6.1 CHEMICAL DIVERSITY: QM9

**Effect of Scaling and Pre-training:** We evaluated the performance of both fine-tuned and from-scratch trained ViSNet and ET models across different capacities, as depicted in Table 2 Figure 10 (Appendix). We assessed model performance on a hold-out test set, leading to the following observations. Firstly, we noted a limited scaling behavior; while moderately larger models generally outperform smaller ones, the performance improvements do not predicatively scale with model size, indicating a manifestation of generalization error due to model over-parameterization. Secondly, models initialized with pre-trained weights consistently outperform their counterparts trained from scratch across both architectures and all model sizes, highlighting the value of the pre-training approach for downstream tasks. Lastly, regardless of the splitting strategy and training approach, ViSNet demonstrates superior performance compared to ET, aligning with the pre-training results shown in Figures 6 and 7 (Appendix).

**Effect of Data Split and Label Uncertainty.** Figure 11 (Appendix) depicts the results from the Random-QM9 experiment. Typically, wider models exhibited improvements in Random-QM9, whereas scaffold-split models struggled, highlighting challenges in domain generalization. Increased model parameters might suggest overfitting rather than true generalization enhancement. Pre-trained models, exposed to a broader molecular structure range in the PCQM dataset (including QM9 molecules), showed superior transferability to downstream tasks, as reflected in the performance disparity between pre-trained and from-scratch models discussed earlier. Additionally, evaluation errors are significantly attributed to label uncertainty. For instance, variations in HOMO-LUMO gap predictions among common DFT methods for three QM9 molecules, detailed in Appendix E and Zhao & Truhlar (2005); Zhang & Musgrave (2007); Mardirossian & Head-Gordon (2017), are markedly greater than the prediction errors, illustrating the inherent uncertainty in training and evaluation labels. And prediction error of B3LYP functional (Lee et al., 1988) used in QM9 dataset are usually at eV-scale. This error in "ground truth" complicates model performance assessment and generalization interpretation, suggesting that training data uncertainty can largely contribute to the observed limitations in model scaling and OOD performance, beyond model limitations.

## 6.2 CONFORMATIONAL DIVERSITY: XXMD

Building upon our analysis of chemical diversity, we now focus on conformational diversity within a single stilbene molecule using the Random-xxMD. We begin our analysis by controlling the number of samples used to supervise the network. We prepared models with varying widths and training strategies to examine dataset and parameter scaling effects, as well as the transferability of embeddings learned from equilibrium and non-equilibrium structures. As illustrated in Figures 12 (Appendix), Our findings highlight three key trends: (1) Pre-training shows diminishing returns beyond a certain dataset size, suggesting a nuanced balance between pre-training depth and dataset scale. (2) In data-rich scenarios, larger models consistently outperform. (3) In data-limited situations, only models pre-trained on the equilibrium PCQM dataset effectively enhance performance. These observations echo the phenomenon of "parameter ossification" mentioned in pre-training literature (Hernandez et al., 2021), where pre-trained models exhibit early performance saturation. Chen et al. (2024) discussed this issue for GIN models pretrained on molecular datasets at the $10^5$ sample scale. Our findings extend this concept to Geom-GNNs, demonstrating that ossification can occur even at the $10^3$ sample scale. Notably, we did not observe ossification effects in the QM9 dataset, despite its orders-of-magnitude-more samples, suggesting that ossification is related to both the number and the correlation among samples.

Table 3: Performance of models with varying width and training strategy on xxMD-Temporal subsets. Best models are picked based on the force regression in terms of MAE. Only models pre-trained on Denali dataset can positively transfer, while ET receives more benefits from pre-training comparing to ViSNet.

| Subset | Best Dim | Model | Energy (meV) | Force (meV/Å) |
|---|---|---|---|---|
| | | **ViSNet** | | |
| Stilbene | 128 | PT-Denali | 321 | 137 |
| Azobenzene | 256 | No PT | 75 | 72 |
| Malonaldehyde | 256 | PT-Denali | 95 | 142 |
| Dithiopehene | 256 | No PT | 87 | 55 |
| | | **ET** | | |
| Stilbene | 384 | PT-Denali | 348 | 142 |
| Azobenzene | 64 | PT-Denali | 127 | 98 |
| Malonaldehyde | 384 | PT-Denali | 144 | 157 |
| Dithiopehene | 64 | PT-Denali | 78 | 74 |

Next, we examined the performance on the Temporal-xxMD dataset, using all four systems: azobenzene, stilbene, malonaldehyde, and dithiophene. Figures 13 (Appendix) illustrates the results for energy and force regressions. The trends observed in this experiment differ notably from our earlier findings: (1) pre-trained embeddings from the Denali dataset demonstrate superior transfer compared to those from PCQM. Table 3 suggests that exposure to non-equilibrium conformations during pre-training enhances the model's ability to generalize to the more diverse conformational space in the xxMD dataset. When both the pre-training and downstream datasets lack coverage of non-equilibrium conformations, the network struggles to extrapolate effectively. However, networks that have learned non-equilibrium representations from the Denali dataset show improved transfer to reactive, highly non-equilibrium conformations. (2) we observe that pre-training does not universally improve performance. Specifically, while ET models consistently benefit from pre-training on either Denali or PCQM datasets, ViSNet models do not always show positive transfer. Greater benefit from pre-training for ET is attributed to its lower baseline performance, allowing more room for improvement since it has not yet reached the underlying DFT uncertainty.

## 7 CONCLUSION

In short, this work explored the application of Geom-GNNs from a new perspective of represenation learning where pre-trained all-atom Geom-GNNs are effective zero-shot transfer learners that can faithfully describe complicated protein environment with all-atom granularity. To study the expressive power of Geom-GNNs with varying architectures and configurations, we characterize the scaling behaviors of Geom-GNNs on self-supervised, supervised and unsupervised tasks. Deviating from common neural scaling power-laws, Geom-GNNs saturate early and are bounded by issues such as under-reaching and over-smoothing. Though there are

benefits of scaling on supervised and unsupervised tasks, other aspects become more crucial, such as addressing data label uncertainty and crafting optimal architectures for protein modeling with pre-trained all-atom embeddings. We hope our work can inspire to rethink how should Geom-GNNs be trained and applied on a broader spectrum of problems. Limited by the scope of the paper, we only explored denoising pre-training objective, while future efforts include expanding the pre-training scope to include molecular topology and exploring co-pre-trained models for applications such as conditional molecule optimization, protein design, and protein-protein interaction.

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

## A    APPENDIX

## B    EQUIVARIANT GRAPH NEURAL NETWORKS

**Equivariant Geom-GNNs** are essential in chemistry and biology for leveraging spatial data like atomic positions while maintaining rotational and translational invariance and equivariance. Equivariance ensures that a function $f$ transforms its output consistently with its input transformation, expressed mathematically as $f(T(x)) = T(f(x))$. Geom-GNNs using invariant scalars (Schütt et al., 2018; Gasteiger et al., 2020) can easily achieve roto-invariance, but they fall short in predicting general equivariant properties other than derivatives with respect to the input coordinates. Group-equivariant Geom-GNNs GNNs (Batzner et al., 2022; Batatia et al., 2022; Musaelian et al., 2023) utilize spherical harmonics and group representation theory. However, these architectures are expressive yet memory-intensive and computationally expensive. The emergence of vector-based equivariant Geom-GNNs GNNs, such as (Schütt et al., 2021; Thölke & De Fabritiis, 2022; Wang et al., 2024), has shown promising results while scaling more efficiently.

## C    NEURAL SCALING LAW ON LANGUAGE

**Neural Scaling Laws** describe empirically-derived power-law relationships between model performance and various scaling factors. These laws have been instrumental in understanding and predicting the behavior of large language models. Kaplan et al. (2020) identified key power-law relationships between the pre-training loss $L$ and several variables: $L \propto N^{-\alpha}, \quad L \propto D^{-\beta}, \quad L \propto C^{-\gamma}$ where $N$ is the number of model parameters, $D$ is the dataset size, and $C$ is the amount of compute. These laws have elucidated performance disparities between different neural architectures, such as Long Short-Term Memory (LSTM) networks (Hochreiter & Schmidhuber, 1997) and Transformers (Vaswani et al., 2017), particularly in terms of pre-training loss. Subsequent work by Hoffmann et al. (2022) and Muennighoff et al. (2024) further refined these relationships, considering factors like optimal batch size, compute-optimal training, and data constraint.

## D    DATASET SUMMARY

Table 4: Overview of datasets used for pretraining and downstream tasks, detailing their purpose, molecular domains, structural types (equilibrium vs. non-equilibrium), labeled properties, the number of samples, and the method of data splitting.

| Purpose | Name | Domain | Structure | Labels | #samples | Split |
|---|---|---|---|---|---|---|
| Pretraining | PCQM4Mv2[†] | Small molecules | Equilibrium | Conformation | 3,378,606 | N/A |
| | OrbNet-Denali[‡] | Molecules complexes | Non-equilibrium | Conformation | 2,383,351 | N/A |
| Downstream | QM9[§] | Small molecules | Equilibrium | Conformation, quantum properties | 116,038 | Scaffold |
| | xxMD[¶] | Small molecules | Non-equilibrium | Conformation, potential, force | Varies | Temporal |
| | Fold[‖] | Proteins | Equilibrium | Conformation | 16,292 | Scaffold |
| | Alanine Dipeptide[**] | Mini peptide | Non-equilibrium | Conformation | 3 x 250 ns trajectories | Random |
| | Pentapeptide[††] | Mini peptide | Non-equilibrium | Conformation | 25 x 500 ns trajectories | Trajectory |
| | λ6-85[‡‡] | Protein | Non-equilibrium | Conformation | 500 x trajectories with varying time | Trajectory |

[†]https://ogb.stanford.edu/docs/lsc/pcqm4mv2/

[‡]https://figshare.com/articles/dataset/OrbNet_Denali_Training_Data/14883867

[§]https://pytorch-geometric.readthedocs.io/en/latest/generated/torch_geometric.datasets.QM9.html

[¶]https://github.com/zpengmei/xxMD

[‖]https://github.com/phermosilla/IEConv_proteins#download-the-preprocessed-datasets

[**]https://markovmodel.github.io/mdshare/ALA2/#alanine-dipeptide

[††]https://markovmodel.github.io/mdshare/pentapeptide/#peptide

[‡‡]https://exhibits.stanford.edu/data/catalog/mh050cw6709

# E UNCERTAINTY IN QM9 LABELS

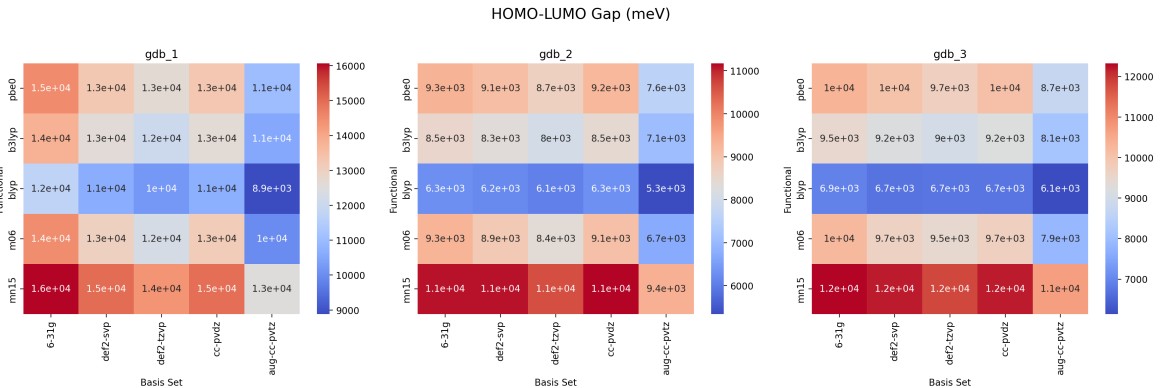

Figure 5: HOMO-LUMO gap of 'gdb1', 'gdb2' and 'gdb3' predicted by varying combinations of common basis sets and density functionals. Different combinations showed considerable variance.

As shown in Figure 5, we scanned combinations of five commonly used functionals (Zhao & Truhlar, 2008; Haoyu et al., 2016; Adamo & Barone, 1999; Lee et al., 1988) and basis sets (Weigend, 2006; Ditchfield et al., 1971; Dunning Jr, 1989), resulting in 25 distinct computational methods using PySCF package (Sun et al., 2020). The standard deviations of the HOMO-LUMO gap predictions were substantial: $12571 \pm 1775$, $8411 \pm 1626$, and $9333 \pm 1758$ meV for gdb1, gdb2, and gdb3, respectively. Such uncertainty from DFT calculations has been widely studied in chemistry community (Zhang & Musgrave, 2007), especially the accuracy of B3LYP functional used in QM9 dataset is non-ideal per current standard (Zhao & Truhlar, 2005; Goerigk & Grimme, 2011; Mardirossian & Head-Gordon, 2017).

## F  HYPERPARAMETERS SUMMARY

Table 5: Hyperparameters used for different models used for pre-training.

| Hyperparameter | ET and ViSNet |
|---|---|
| Hidden dimension | 64,128,256,384,512 |
| # MP layers | 2-10 |
| # Attention heads | 8 |
| # RBFs | 32 |
| Layernorm | Max-Min (ViSNet), Whitened/None (ET) |
| Radius cutoff | 3-9 |
| Epochs | 10 |
| Batch size | 400 |
| Learning rate | 0.0005 |
| Optimizer | Adam |
| Noise level | 0.05 (PCQM), 0.2 (Denali) |

Table 6: Hyperparameters used for different models in the supervised regression experiments using QM9 and xxMD datasets. In xxMD experiments, we set the weight on the force and energy as 100:1.

| Hyperparameter | QM9 | xxMD |
|---|---|---|
| Hidden dimension | 64,128,256,384,512 | 64,128,256,384 |
| # MP layers | 6 | |
| # Attention heads | 8 | |
| # RBFs | 32 | |
| Layernorm | Max-Min, Whitened | Max-Min, None |
| Radius cutoff | 5 | |
| Epochs | 500 | 200 |
| Early Stopping (epochs) | 10 | 20 |
| Batch size | 50 | 12 |
| Learning rate | 5e-5/1e-4/5e-4 (from-scratch) | 1e-4/5e-4 (from-scratch) |
| Optimizer | Adam | |
| Scheduler | ReduceLROnPlateau | |
| Scheduler patience (epoch) | 8 | |

Table 7: Hyperparameters used for different models in the VAMP experiments.

| Hyperparameter | Ala2 | Pentapeptide | $\lambda$6-85 |
|---|---|---|---|
| Base model | | ViSNet (PCQM) | |
| Hidden dimension | | 64,128,256,384 | |
| Output dimension | 6 | 1-10 | 6 |
| # MP layers | | 6 | |
| # Attention heads | | 8 | |
| # RBFs | | 32 | |
| Layernorm | | Max-Min | |
| Radius cutoff | | 5 | |
| Token mixer | None | None | None, MLP-Mixer, Transformer |
| Epochs | 10 | 50 | 50 |
| Early Stopping (train,val batches) | | 100,5 | |
| Batch size | | 5000 | |
| Learning rate | 5E-04 | 2E-04 | 2E-04 |
| Optimizer | | Adam | |

Table 8: Hyperparameters used for fold classification experiments.

| Hyperparameter | ProNet-Amino-Acid/ViSNet |
|---|---|
| Hidden dimension | 64,128,256,384 |
| # MP layers | 2 |
| # Attention heads | 8 |
| # RBFs | 32 |
| Layernorm | Max-Min |
| Radius cutoff | 5 (ViSNet), 10 (ProNet) |
| Epochs | 400 |
| Batch size | 32 |
| Learning rate | 0.0005 |
| Optimizer | Adam |
| Coord noise | TRUE |
| Rep. noise | FALSE |
| Res. Mask | TRUE |
| Dropout | 0.4 (ProNet-AA), 0.5 (ProNet-Vis) |

## G    ILLUSTRATION OF DENOISING RESULTS

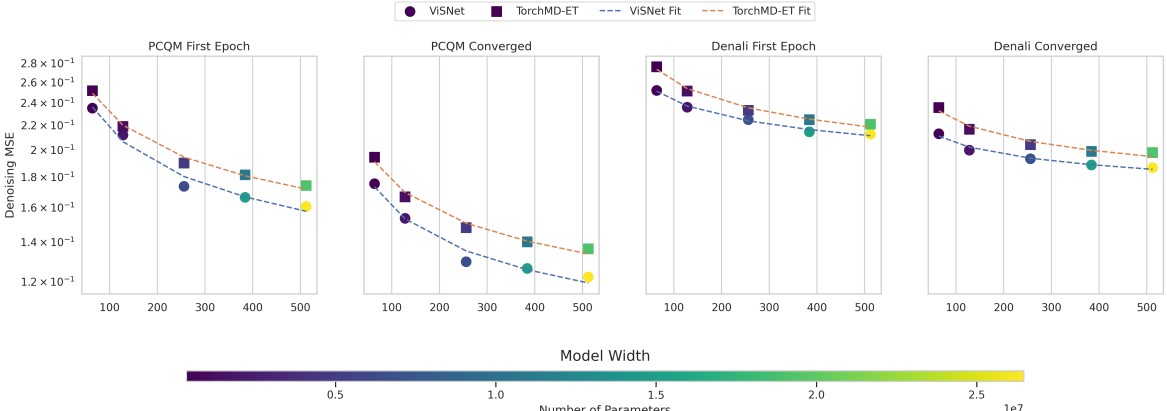

Figure 6: Scaling behavior of ViSNet and ET models on PCQM and Denali datasets. The plots show the denoising Mean Squared Error (MSE) against model width and colored with the number of model parameters. Results are presented for both the first epoch and converged performance on each dataset. Solid lines with markers represent observed data, while dashed lines indicate power-law fits.

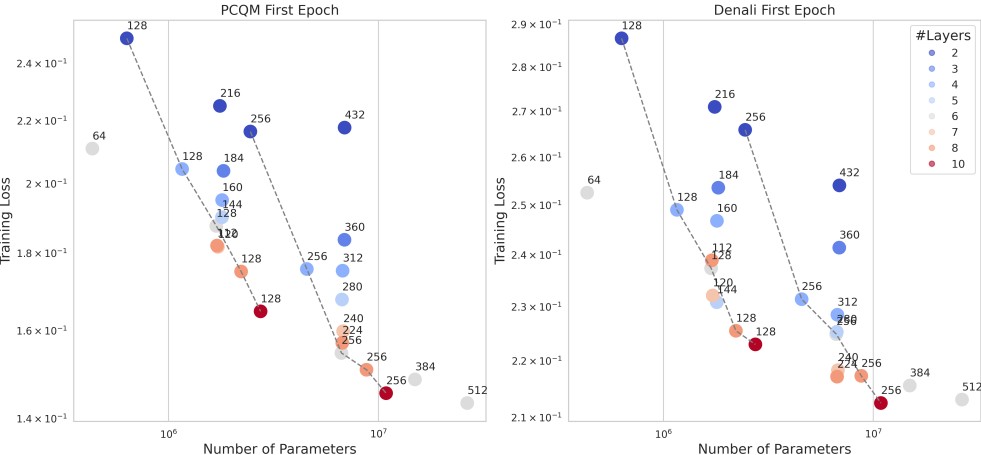

Figure 7: Scaling behavior of ViSNet depth for the first epoch results of training on PCQM and Denali datasets. The plot shows the relationship between the number of layers and the pre-training loss, illustrating the initial rapid improvement followed by diminishing returns as depth increases.

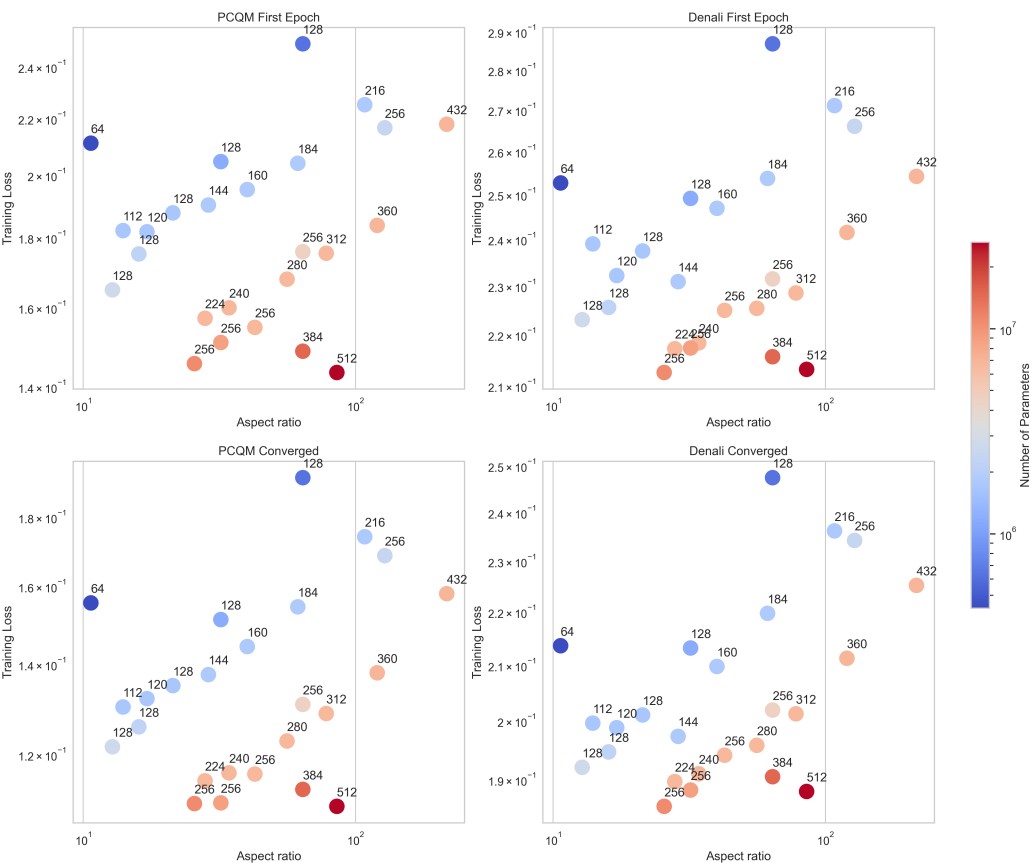

Figure 8: Influence of the aspect ratio on the self-supervised denoising pre-training task using the ViSNet model. With similar amount of parameters, pre-training performance improves as the aspect ratio decreases.

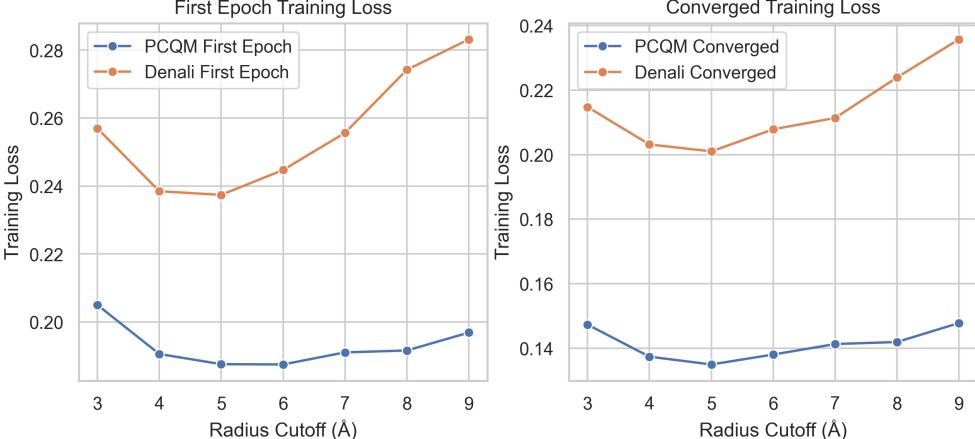

Figure 9: Influence of the radius cutoff distance on the self-supervised denoising pre-training task using the ViSNet model. Orange and blue curve refer to models trained using Denali and PCQM datasets, respectively. Both the first epoch and the converged loss are reported. Models pre-trained on both datasets demonstrate same optimal radius cutoff at 5 Å. Deviating further from the optimal cutoff, the penalty of models pre-trained on Denali dataset is more severe than the PCQM.

# H ILLUSTRATION OF QM9 RESULTS

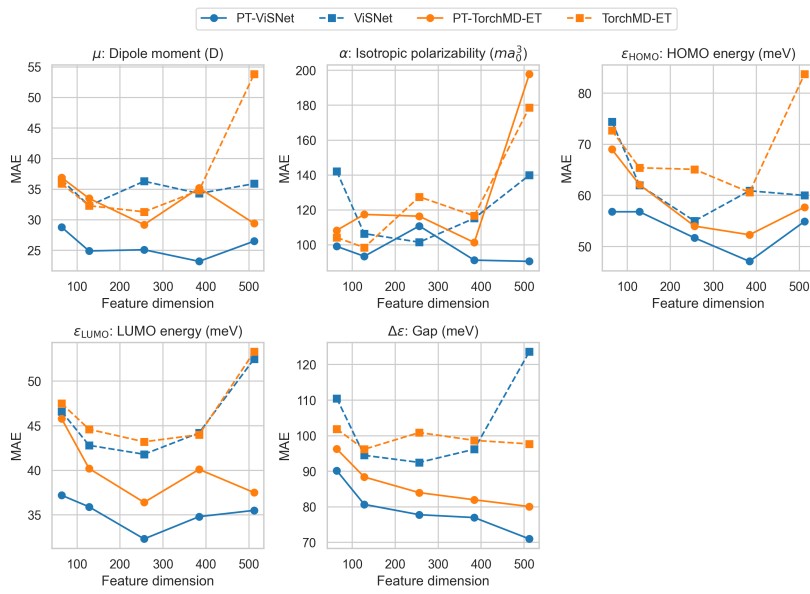

Figure 10: Performance comparison of ViSNet and ET models on Scaffold-QM9 dataset with varying model widths. The plot shows Mean Absolute Error (MAE) against model width, comparing pre-trained (fine-tuned) models with those trained from scratch. Dashed lines refer to non pre-trained models and solid lines refer to pre-trained models using PCQM dataset. ViSNet consistently outperforms ET. Pre-trained models show better performance than their from-scratch counterparts.

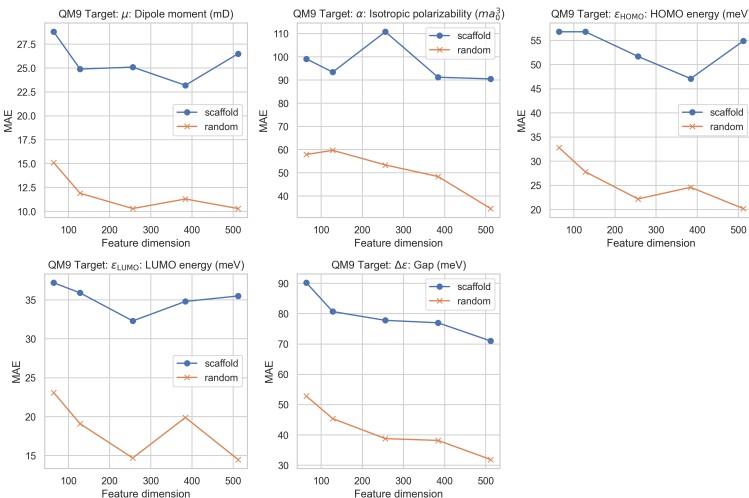

Figure 11: Illustration of the performance of PCQM pre-trained ViSNet models with varying width on Random and Scaffold-QM9. The plots indicate different scaling behaviors with respect to the model parameters. Random-QM9 can always benefit from scaling the model size while the Scaffold-QM9 cannot.

# I ILLUSTRATION OF XXMD RESULTS

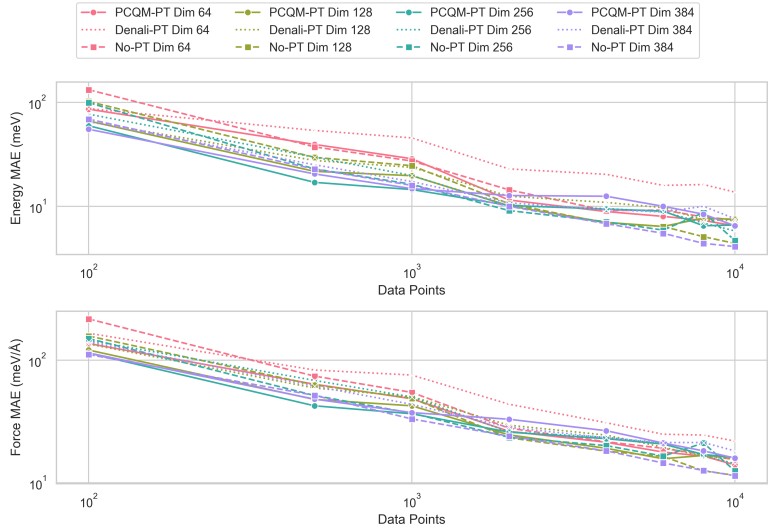

Figure 12: Performance comparison of ViSNet models on the randomly split stilbene subset of xxMD-DFT for energy and force prediction. The plot shows the Mean Absolute Error (MAE) of potential energy against the number of training samples for models with varying widths [64, 128, 256, 384]. Three training scenarios are compared: from-scratch, pre-trained on PCQM (Equilibrium structures), and pre-trained on Denali (Non-equilibrium structures). This experiment illustrates: (1) we observe an **intersection point** beyond $10^3$ samples which pre-training may actually hinder performance compared to training from scratch. This suggests a complex relationship between pre-training benefits and downstream dataset size. (2) in data-abundant regimes (with more than $10^3$ samples), larger models generally demonstrate superior performance, aligning with typical scaling laws in machine learning. (3) in data-sparse scenarios, only models pre-trained on the PCQM dataset show positive transfer to downstream tasks, highlighting the difference of learned embedding from equilibrium and non-equilibrium pre-training datasets.

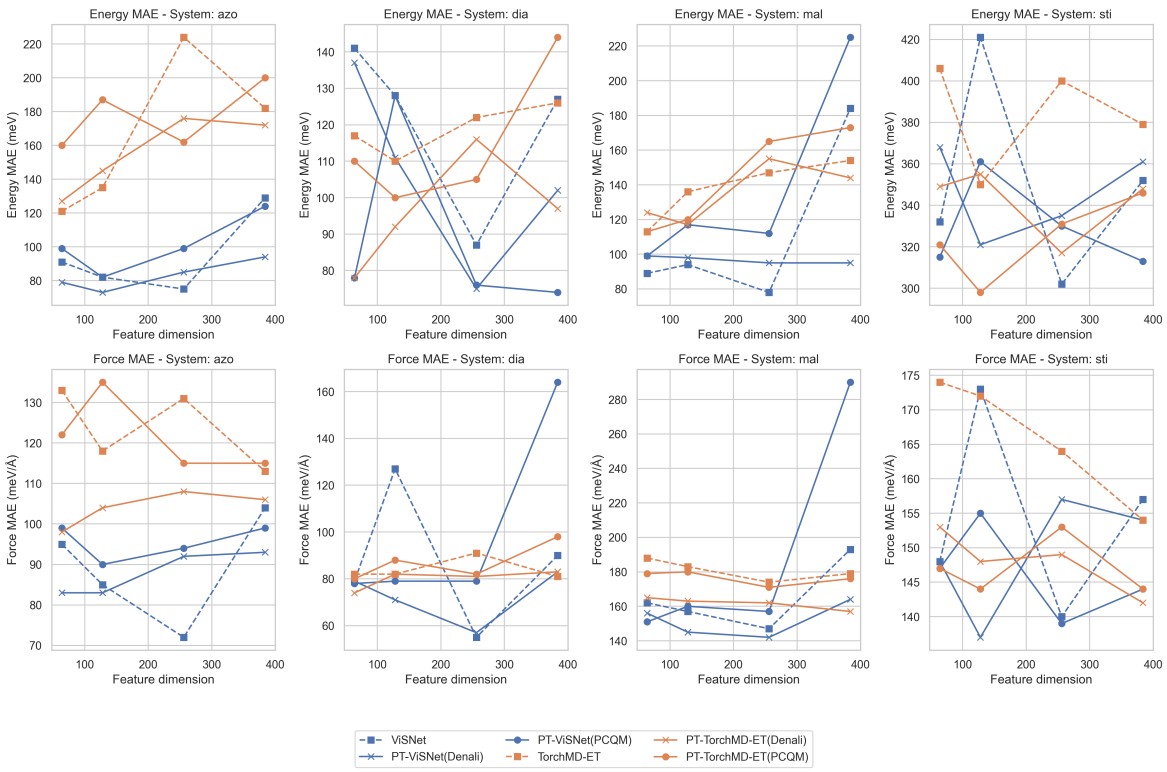

Figure 13: Comparison of Energy MAE and Force MAE using Temporal-xxMD-DFT dataset across different systems for various architectures and training configurations. Each subplot represents a different system ('azo': azobenzene, 'sti': stilbene, 'mal': malonaldehyde, and 'dia': dithiophene), with consistent color coding for architectures and distinct markers indicating different training methods and pre-training datasets. Top row shows Energy MAE, while the bottom row shows Force MAE, demonstrating model performance variation with feature dimensions.

## J    VAMPNET OBJECTIVE AND ILLUSTRATION OF MORE VAMP RESULTS

VAMPNet is a deep-learning-based objective that learns the projection from high-dimensional feature space to low-dimensional latent space while preserving the slow dynamics of a given time series data. Consider a molecular system characterized by a high-dimensional configuration space, $\mathcal{X}$. Let $\{x_t\}_{t=1}^T$ denote a trajectory from a molecular dynamics simulation, where $x_t \in \mathcal{X}$ represents the molecular configuration at time $t$. For general non-stationary and non-revsersible dynamics, the Koopman operator $\mathcal{K}_\tau$ describes the time evolution of an observable $\psi(x)$ over a time lag $\tau$:

$$\mathcal{K}_\tau \psi(x_t) = \mathbb{E}[\psi(x_{t+\tau}) \mid x_t].$$

To approximate the Koopman operator in practice, we used linear superposition ansatz as a finite set of basis functions (features).

$$f(x) \approx \sum_{i=1}^N c_i \chi_i(x),$$

where $c_i$ are coefficients, and $\chi_i(x)$ are basis functions or features that depend on the conformational degrees of freedom of the system. Then, the finite-dimensional approximation of the Koopman matrix $K$ is defined as:

$$K = C_{00}^{-1} C_{01}.$$

where the instantaneous covariance matrices $C_{00}$ and $C_{11}$, as well as the time-lagged covariance matrix $C_{01}$, are computed as follows:

$$C_{00} := \frac{1}{T - \tau} \sum_{t=0}^{T-\tau} (\chi(t) - \mu_0)(\chi(t) - \mu_0)^\top$$

$$C_{11} := \frac{1}{T - \tau} \sum_{t=\tau}^{T} (\chi(t) - \mu_1)(\chi(t) - \mu_1)^\top$$

$$C_{01} := \frac{1}{T - \tau} \sum_{t=0}^{T-\tau} (\chi(t) - \mu_0)(\chi(t+\tau) - \mu_1)^\top$$

The mean vectors $\mu_0$ and $\mu_1$ are computed from all data excluding the last $\tau$ steps and the first $\tau$ steps of every trajectory, respectively:

$$\mu_0 := \frac{1}{T - \tau} \sum_{t=0}^{T-\tau} \chi(t)$$

$$\mu_1 := \frac{1}{T - \tau} \sum_{t=\tau}^{T} \chi(t)$$

To refine this approximation, the half-weighted Koopman matrix $\bar{K}$ is introduced, which normalizes the covariance matrices using their inverse square roots:

$$\bar{K} = C_{00}^{-1/2} C_{01} C_{11}^{-1/2}.$$

The matrix $\bar{K}$ is significant as it encodes the optimal reduced model for the dynamical system, particularly preserving the slowest modes. The singular value decomposition (SVD) of $\bar{K}$ is then performed to extract the primary components of the dynamics:

$$\bar{K} = U'SV'^\top,$$

where $U'$ and $V'$ represent the left and right singular vector matrices, and $S$ is the diagonal matrix of singular values.

To derive the singular functions, the input feature vectors are subsequently projected onto the singular vectors, yielding the left singular functions $\psi(t)$ and right singular functions $\phi(t)$:

$$\psi(t) = U'^\top C_{00}^{-1/2}(\chi(t) - \mu_0), \quad \phi(t) = V'^\top C_{11}^{-1/2}(\chi(t) - \mu_1).$$

The VAMP-2 score, which is a measure of the quality of the dynamical model in preserving the slow dynamics, is directly related to the singular values of the half-weighted Koopman matrix $\bar{K}$:

$$\text{VAMP-2} = \|C_{00}^{-1/2} C_{01} C_{11}^{-1/2}\|_F^2 = \sum_{i=1}^{d} \sigma_i^2(\bar{K}),$$

where $\|\cdot\|_F$ denotes the Frobenius norm, and $\sigma_i(\bar{K})$ are the singular values of the half-weighted Koopman matrix $\bar{K}$. Then the VAMP-2 score can be used to optimize the network, which is equivalent to finding a representation $\psi$ such that the corresponding Koopman matrix captures the dominant dynamical modes.

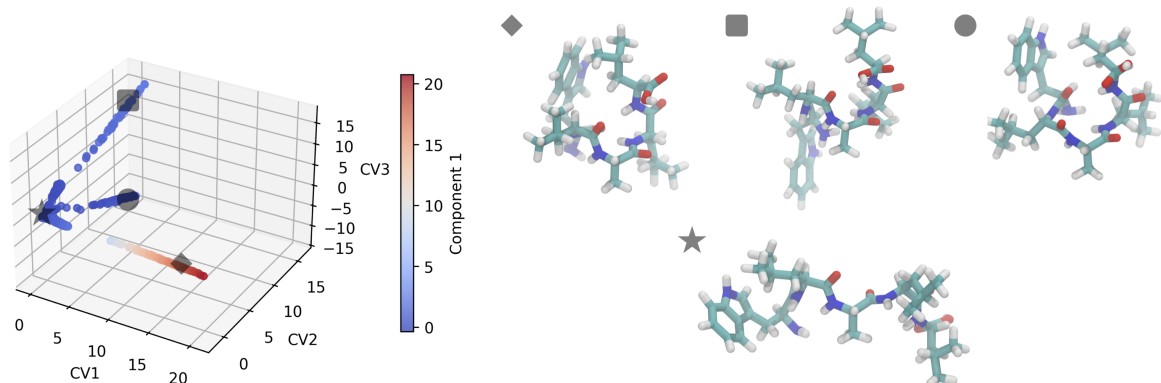

Figure 14: Characteristic structures of pentapeptide in the trajectories visualized on the three-dimensional projection of the learned 6-state VAMP coordinates.

## K    ILLUSTRATION OF FOLD CLASSIFICATION RESULTS AND EMBEDDINGS

In this study, we explore whether graph embeddings can effectively represent local all-atom structures. To achieve this, we fragmented the proteins into individual residues while retaining a sliding window of 3/2 residues based on the sequence to provide contextual information. We then input the corresponding atomic coordinates into the Geom-GNNs to obtain the embedding feature vectors. This approach also helps mitigate the over-smoothing problem, where global information is mistakenly mixed into local features, thus deteriorating the quality of local embeddings.

To visualize the features learned by Geom-GNNs, we generated two-dimensional t-SNE embeddings of the latent space for both pre-trained and untrained networks using the validation set of the HomologyTAPE dataset, as shown in Figures 17 and 18. While equivariant features provide higher resolution, they are often challenging to reduce in dimensionality with standard techniques like principal component analysis (PCA) or t-SNE. To enhance visualization, we regulated the scalar channels of the network during denoising pre-training, following the approach of Liu et al. (2022). Additionally, we noticed multiple preprocessing errors in the provided dataset when visualizing characteristic structures in the graph embedding space in Figure 19. We followed the Wang et al. (2022) to use the pre-processed datasets from Hermosilla et al. (2021) (available at `https://github.com/phermosilla/IEConv_proteins`). These preprocessing errors are difficult to detect in feature spaces defined by dihedral and torsion angles, but are more apparent in the graph embedding space, which could negatively affect regression accuracy.

It is important to note that projections based on dihedral and torsion angles should not be directly compared with graph embeddings. While these angular features are effective for characterizing secondary structures, they lack all-atom resolution, are coarse-grained, and do not account for environmental effects. It is still a missing piece of study to define some general emperical descriptors that can roughly describe the interactions between residues and their side-chains. Our comparisons serve to confirm consistency and agreement between the different representations. Interestingly, even untrained Geom-GNNs are able to capture patterns in the dataset, similar to untrained convolutional neural networks (CNNs) in vision tasks, as demonstrated by Ulyanov et al. (2018). However, the features derived from untrained networks are generally sparse and less interpretable.

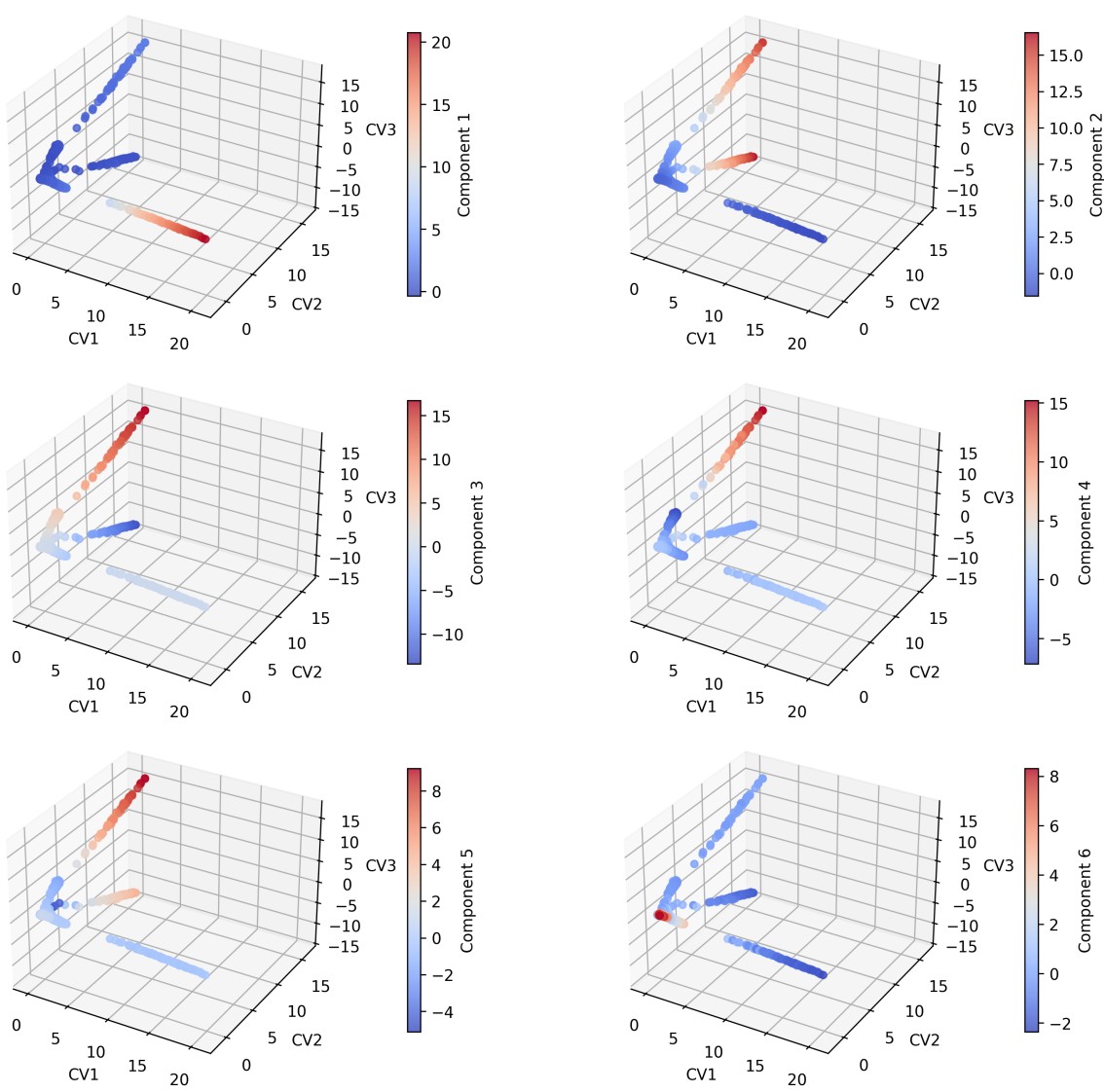

Figure 15: Three-dimensional projections of learned singular vectors in descending order from the 6-state VAMP model. Lag time: 0.5 ns.

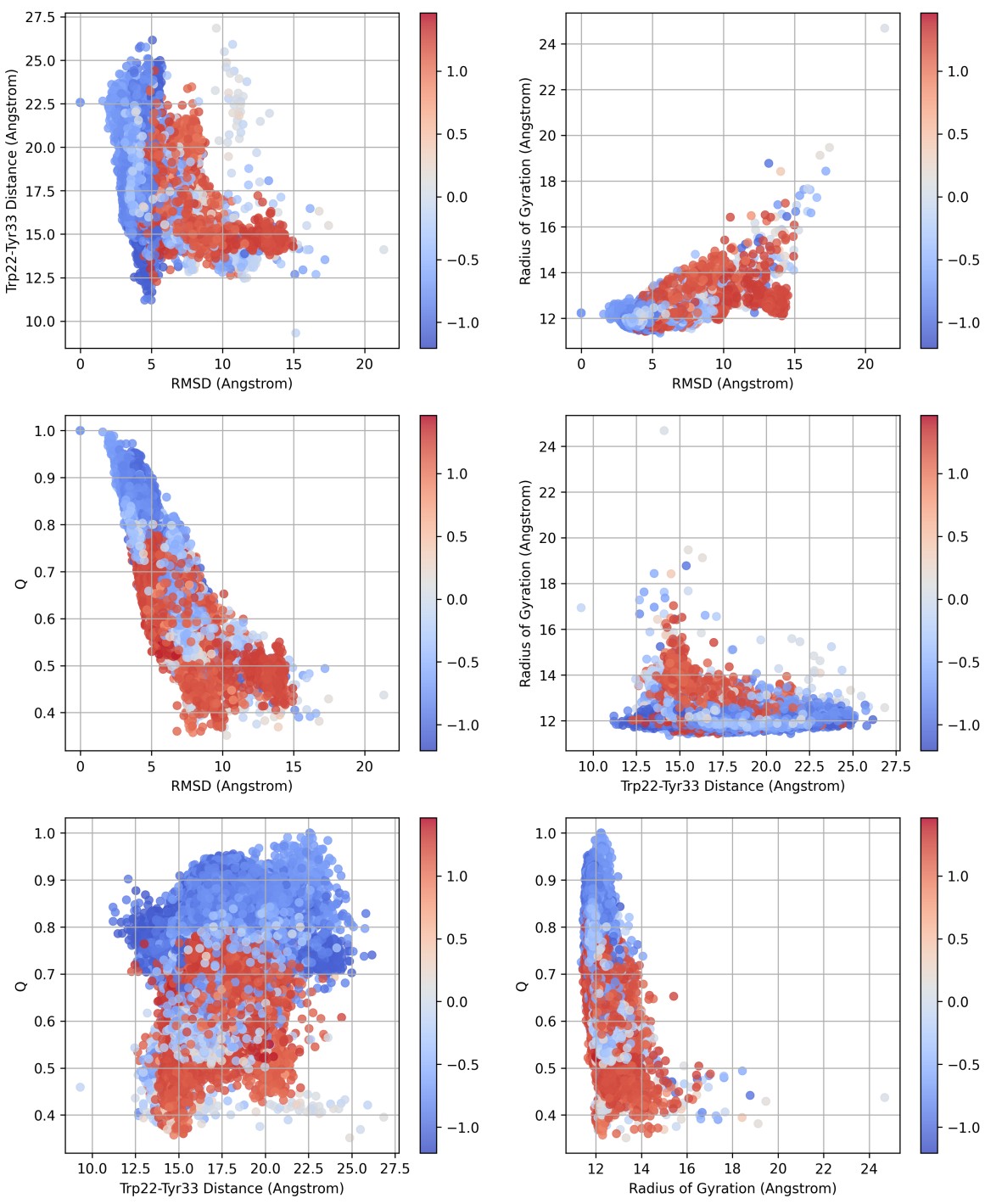

Figure 16: Projection of the first VAMP singular vector (Lag time: 50 ns) on typical collective variables such as radius of gyration, structural rooted mean squared difference (RMSD) to the crystal structure, fraction of native contacts (Q), pair-wise distance of Trp22 and Tyr33. Bowman et al. (2011) indicates 2-3 states on those common collective variables, and the results are comparable. We randomly drew 500 trajectories out of the original dataset to train the VAMP head.

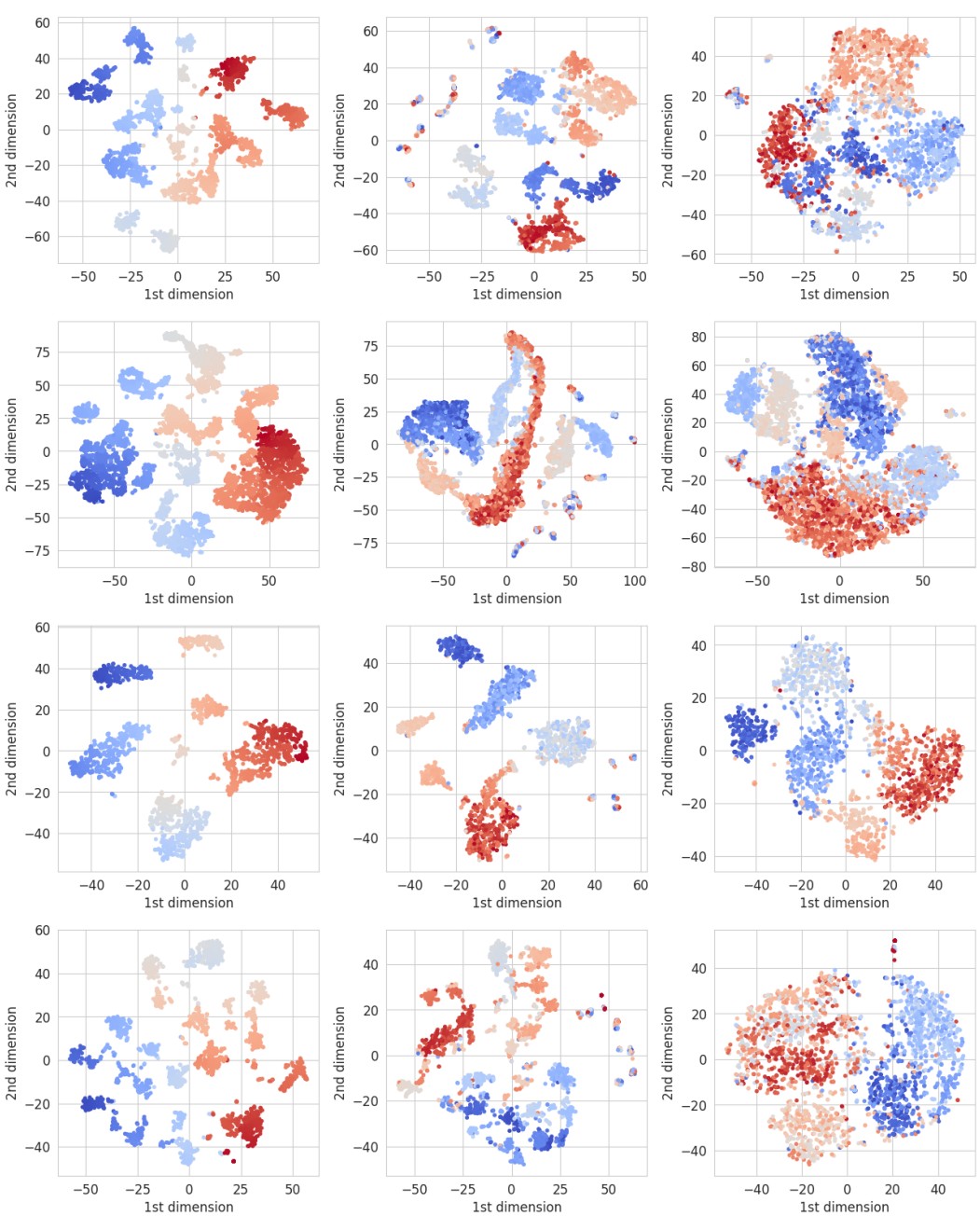

Figure 17: Two-dimensional t-SNE visualization of residues in the validation subset of HomologyTAPE dataset for the folding classification task. Points are colored according to the one-dimensional t-SNE projection of the angles. Columns represent: (Left) backbone dihedral and sidechain torsion angles, (Middle) pre-trained graph embedding from ViSNet with a width of 128, and (Right) untrained graph embedding from the same model. Rows correspond to different amino acids: (1) Histidine (His), (2) Aspartic Acid (Asp), (3) Cysteine (Cys), and (4) Methionine (Met).

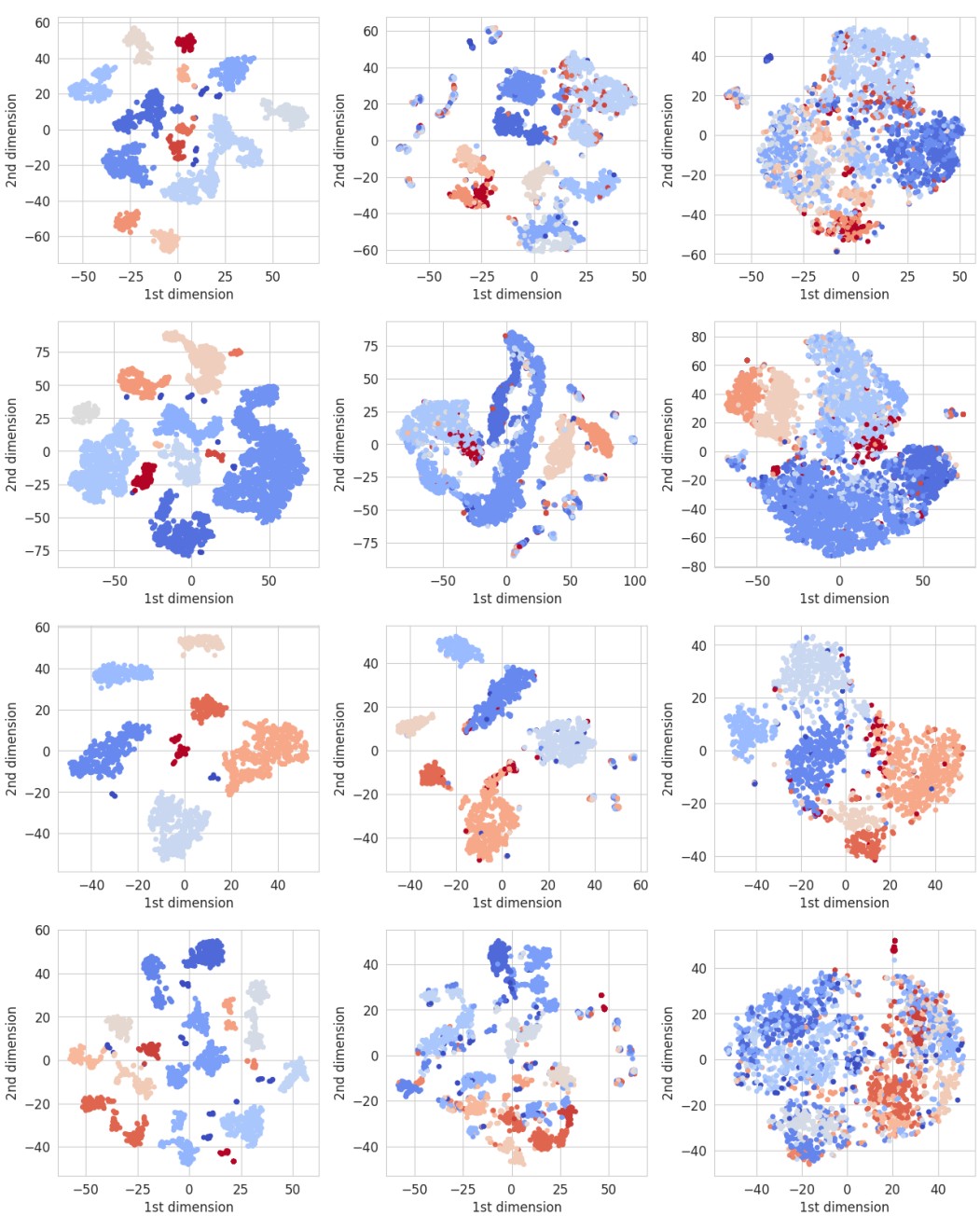

Figure 18: Two-dimensional t-SNE visualization of residues in the validation subset of HomologyTAPE dataset for the folding classification task. Points are colored according to DBSCAN clustering labels of the angles. Columns represent: (Left) backbone dihedral and sidechain torsion angles, (Middle) pre-trained graph embedding from ViSNet with a width of 128, and (Right) untrained graph embedding from the same model. Rows correspond to different amino acids: ((1) Histidine (His), (2) Aspartic Acid (Asp), (3) Cysteine (Cys), and (4) Methionine (Met).

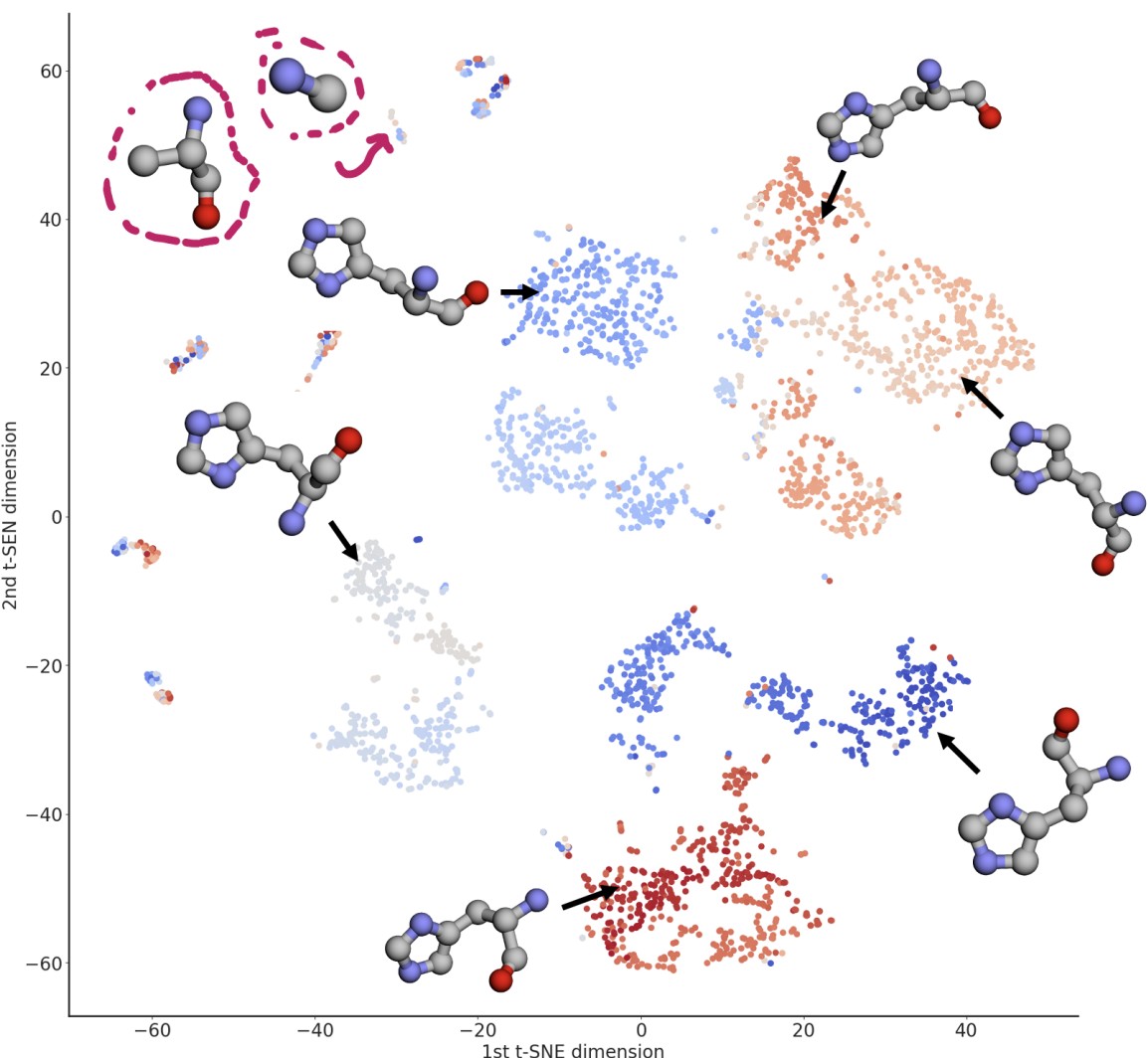

Figure 19: Visualization of characteristic structures for histidine (His) residues using two-dimensional t-SNE projections of pre-trained graph embeddings. The plot shows that structurally similar residues are clustered together, while multiple outliers, likely due to preprocessing errors, are also present. Similar preprocessing errors were observed for other types of residues as well.

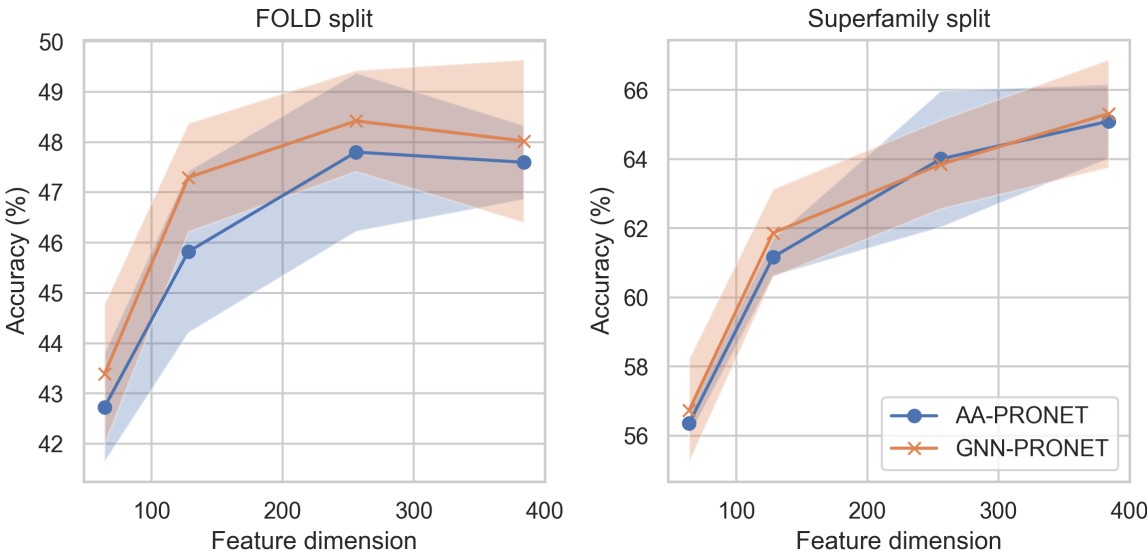

Figure 20: Folding classification accuracy with the fold and superfamily split. For pronet architecture, we restrict ourselves to use amino-acid-level features (AA-PRONET), whereas we supplement the original architecture with the pre-trained GNN features for transfer (GNN-PRONET). It is important to note that our preliminary implementation is not compatible with the data augmentation techniques employed in the original ProNet paper. Our primary objective is not to establish a new state-of-the-art model, but rather to analyze the transferability of pre-trained all-atom graph representations to a distinct architectural framework and task domain. We reproduced the original ProNet results with multiple runs to establish a baseline for comparison. There are minor discrepancies between our reproduced values and those reported in the original paper, presumably due to normal experimental variance with different hyperparameters. As illustrated in plots, both models stopped to improve at 256 width on the fold split, while increasing the model size can continue benefit the superfamily split.

## L    ADDITIONAL RESULTS OF XXMD-TEMPORAL AND QM9 BENCHMARKS

Table 9: Performance comparison across different dimensions and training setups for ViSNet and ET models on four subsets of xxMD-DFT using temporal split.

| Molecule | Model | Dim | No PT | | PT on Denali | | PT on PCQM | |
|---|---|---|---|---|---|---|---|---|
| | | | E | F | E | F | E | F |
| Azobenzene | ViSNet | 64 | 91 | 95 | 79 | 83 | 99 | 99 |
| | | 128 | 82 | 85 | 73 | 83 | 82 | 90 |
| | | 256 | 75 | **72** | 85 | 92 | 99 | 94 |
| | | 384 | 129 | 104 | 94 | 93 | 124 | 99 |
| | ET | 64 | 121 | 133 | 127 | **98** | 160 | 122 |
| | | 128 | 135 | 118 | 145 | 104 | 187 | 135 |
| | | 256 | 224 | 131 | 176 | 108 | 162 | 115 |
| | | 384 | 182 | 113 | 172 | 106 | 200 | 115 |
| Stilbene | ViSNet | 64 | 332 | 148 | 368 | 148 | 315 | 147 |
| | | 128 | 421 | 173 | 321 | **137** | 361 | 155 |
| | | 256 | 302 | 140 | 335 | 157 | 330 | 139 |
| | | 384 | 352 | 157 | 361 | 154 | 313 | 144 |
| | ET | 64 | 406 | 174 | 349 | 153 | 321 | 147 |
| | | 128 | 350 | 172 | 355 | 148 | 298 | 144 |
| | | 256 | 400 | 164 | 317 | 149 | 331 | 153 |
| | | 384 | 379 | 154 | 348 | **142** | 346 | 144 |
| Malonaldehyde | ViSNet | 64 | 89 | 162 | 99 | 156 | 99 | 151 |
| | | 128 | 94 | 157 | 98 | 145 | 117 | 160 |
| | | 256 | 78 | 147 | 95 | **142** | 112 | 157 |
| | | 384 | 184 | 193 | 95 | 164 | 225 | 290 |
| | ET | 64 | 113 | 188 | 124 | 165 | 113 | 179 |
| | | 128 | 136 | 183 | 117 | 163 | 120 | 180 |
| | | 256 | 147 | 174 | 155 | 162 | 165 | 171 |
| | | 384 | 154 | 179 | 144 | **157** | 173 | 176 |
| Dithiopehene | ViSNet | 64 | 141 | 79 | 137 | 79 | 78 | 78 |
| | | 128 | 128 | 127 | 111 | 71 | 128 | 79 |
| | | 256 | 87 | **55** | 75 | 57 | 76 | 79 |
| | | 384 | 127 | 90 | 102 | 83 | 74 | 164 |
| | ET | 64 | 117 | 82 | 78 | **74** | 110 | 80 |
| | | 128 | 110 | 82 | 92 | 82 | 100 | 88 |
| | | 256 | 122 | 91 | 116 | 81 | 105 | 82 |
| | | 384 | 126 | 81 | 97 | 83 | 144 | 98 |

## M    GENERAL RECOMMENDATIONS FOR SCALING AND EVALUATING GEOMETRIC GNNs

To facilitate the training and evaluation of geometric GNNs, we provide an integrated summary of the observations drawn from experiments on self-supervised pre-training, unsupervised transfer, and supervised fine-tuning. These findings offer practical guidance for practitioners aiming to scale and optimize GNN architectures effectively.

From the perspective of pre-training, we observed a consistent decrease in training loss as the depth of the network increased. However, these benefits diminished once a critical depth was reached. For instance, a ViSNet with six layers achieved a training loss comparable to that of an eight-layer ViSNet with the parameter count matched. This plateau effect suggests that, while increasing depth initially enhances the network's learning capacity, over-scaling yields diminishing returns. In molecular property prediction tasks, as demonstrated in

Table 10: Comparison of $\epsilon_{\text{LUMO}}$ and $\Delta\epsilon$ tasks in Scaffold-QM9 across different Layer-Dimensions with various pre-training setups.

| Layer-Dim | Setup | Metrics | |
|---|---|---|---|
| | | $\epsilon_{\text{LUMO}}$ | $\Delta\epsilon$ |
| 6L-128 | No PT | 42.8 | 94.5 |
| | PT on PCQM | 35.9 | 80.7 |
| 5L-144 | No PT | 46.1 | 104.8 |
| | PT on PCQM | 39.5 | 88.3 |
| 4L-160 | No PT | 45.6 | 98.5 |
| | PT on PCQM | 39.8 | 96.7 |
| 3L-184 | No PT | 44.5 | 98.4 |
| | PT on PCQM | 44.1 | 103.0 |
| 2L-216 | No PT | 43.5 | 97.6 |
| | PT on PCQM | 46.6 | 96.0 |

Table 10, increased depth consistently improved prediction accuracy, reaffirming its importance for certain downstream tasks. Nonetheless, in force field prediction tasks, as highlighted by Li et al. (2024) (Figure 2), the advantage of scaling depth similarly diminished beyond a certain point, albeit with a less controlled parameter setup. An intriguing phenomenon emerged concerning the radius graph cutoff. Increasing the cutoff distance initially improved performance but eventually led to deterioration, forming a characteristic U-shaped performance curve (Figure 9). This aligns with findings from Li et al. (2024), suggesting that excessively large cutoff radii may introduce noise from irrelevant long-range interactions, thereby reducing model effectiveness.

In terms of parameter scaling, models with a width of 256 or 384 provided an optimal balance between computational cost and performance across various tasks, as illustrated in Figures 10, 11, 12, 13, and 20. Wider models generally showed only marginal gains relative to their increased computational overhead, indicating diminishing returns for excessive parameter expansion. It is critical to carefully consider the dataset splitting strategy when evaluating the resulting models. Improper splits can lead to overfitting and hinder the generalization of learned representations. For example, scaffold splits are more reliable than random splits for assessing the transferability of molecular representations. Moreover, architecture-level improvements remain necessary. For protein kinetic modeling, incorporating additional Transformer or MLP-Mixer layers significantly improved performance (Table 1). Similarly, in force field prediction tasks involving larger molecules, adding long-range interaction modules further enhanced model predictions if applicable, as evidenced by Li et al. (2024).

