# OpenReview forum: "Pushing the Limits of All-Atom Geometric Graph Neural Networks: Pre-Training, Scaling, and Zero-Shot Transfer"
_ICLR.cc/2025/Conference — ICLR 2025 Poster_

### Official Review · Reviewer_5LXW · 2024-10-26

**Soundness:** 2
**Presentation:** 1
**Contribution:** 1
**Rating:** 6
**Confidence:** 3

**Summary:**

In their work the authors empirically studied properties of pre-trained geometric GNNs (i.e., GNNs with coordinates attached).
They especially seemed to consider pretraining via denoising as introduced by Zaidi et al. (2022).
The authors consider several downstream tasks: molecular dynamics (kinetics modeling), fold classification, prediction of quantum-chemical properties, and, molecule conformations
They study properties such as power-law scaling and find that the geomtric GNNs do not follow that on the pre-training task.
The authors conclude that geometric GNNs "saturate early and are bounded by issues such as under-reaching and over-smoothing" and further that "other aspects become more crucial, such as addressing data label uncertainty and employing active token mixing to mitigate the information bottleneck."

**Strengths:**

relevant research topics in the paper:
- pretraining of (geometric) GNNs
- scaling behavior of GNNs
- oversmoothing, underreaching

experiments:
- experiments considered not only random splitting, but also scaffold and temporal splits

**Weaknesses:**

In general the largest problem to my opinion is that the authors lack to specify a clear research goal. Methodologically there seems not too much novelty.
But if it is primarily an empirical research paper it seems very important to me that the research goal is clearly defined and dependent on that authors have
to reason why they are selecting the problems they look at and why they are selecting the methods they compare to. If a new benchmark dataset is used (as e.g., scaffold QM9), then empirical evaluations/comparisons should take into account at least some previously suggested methods (which are ideally complementary to each other to better see the potentials of each of the methods on the new benchmark and being able to compare to what authors say).
\
\
Details:
\
\
**hard to see the novelty in the paper**:\
Fact that pretraining is useful was already found e.g. by Zaidi et al. (2022).\
No new methodology seems to be suggested (or is "token mixing" the new method)?\
Study of scaling behavior might be interesting and to a certain degree novel for geometric GNNs, but no follow-up investigations seemed to be employed to draw formal connections to GNN properties like oversmoothing, underreaching, etc.
\
\
**paper is structured in a strange way, which makes it hard to follow the paper**:\
It is actually hard to understand what the research aim of the authors was.\
chapter 4, 5, and, 6, actually seem to be about empiricial experiment results.\
The setups are partly however already explained in chapter 3.\
chapter 4 and 6 show performances on problems\
chapter 5 however studies power-law behavior and other ablations.\
In sum it gets hard to follow the paper. Better reasoning why some experiments are applied at all (with respect to the general research goal) and why they are done as they are done is necessary (e.g., why which methods are selected for comparison or why which dataset is used).
\
\
**experiments**:\
although there are some good points as mentioned in strengths (such as the splits), it will get hard to understand how large the impact of pretraining really is, as the authors only test very few method once with pretraining and once without pretraining.\
\
also the there is no good argument in the paper, why the authors exactly compare to those methods they selected to compare to
e.g.:\
for molecular dynamics, why don't they compare to Timewarp\
for QM9 and xxMD they could e.g. compare to "E(n) Equivariant Graph Neural Networks", "SE(3)-Transformers", DimeNet++, MACE, etc.\
An option to get more impression on the significance of the author's results would be to additionally compare to standard QM9, etc. (where there are also a lot of method comparisons out).
\
\
**minor points**:\
"token mixing" not defined, but heavily used\
grammatical errors/typos: "In silico molecular computation and simulation are indispensable tool in..."

**Questions:**

- Why is the VAMP task considered zero-shot?
- Appendix D: Why is there a difference between datasets for pretraining and downstream? According to Table 2, QM9 there seem also experiments with pretraining. Are models in Table 3 not fine-tuned?

---

> ### Author Response · Authors · 2024-11-20
> **Response to Reviewer 5LXW Part 1**
>
> ## Reply to Summary
>
> We sincerely appreciate the reviewer **i3nd** for the feedback and comments to improve the quality of the paper. Before proceeding further, we think it is necessary to reformulate the summary to reiterate the motivation of our paper. Instead of extending denoising pre-training techniques, investigating the pre-training task choice, or developing model architectures, our paper aims to answer a previously unanswered research question in graph representational learning, which fits the ICLR venue:
> **"Are pre-trained all-atom geometric graph neural network (GNN) representations transferable to protein modeling, and how expressive are they?"**
>
> To answer this research question, our contributions are as follows:
> 1. **Scaling Behaviors of Geometric GNNs:**
>    We studied the scaling behaviors of state-of-the-art geometric GNNs in unsupervised, self-supervised, and supervised setups, instead of focusing on developing new model architectures or pre-training objectives.
> 2. **Demonstrating Transferability:**
>    We pre-trained these GNNs on small molecular datasets and demonstrated their transferability to proteins with all-atom resolution, highlighting their expressiveness in these settings.
>
> ### Pre-training and Zero-Shot Transfer Learning
>
> In terms of pre-training, we studied scaling behaviors with various model configurations. In the zero-shot transfer learning setup, we inferred the atomistic embeddings of all-atom peptides and proteins, coarse-grained to residue-wise embeddings, and organically combined them with other architectures for conformational kinetic modeling (VAMPNet) and fold classification tasks. In both tasks, pre-trained all-atom embeddings demonstrated excellent transferability.
>
> ### Small Molecule Setups
>
> In the small molecule setups, we studied molecular property prediction (QM9) and molecular force field prediction (xxMD) with both pre-trained and non-pre-trained models. In all the aforementioned setups, we did not find predictable power-law scaling (not just on the pre-training task). During pre-training, we observed that shallow models exhibit much higher pre-training loss compared to deeper models with similar parameter counts (hence, why we mentioned **under-reaching**). Additionally, we found that the benefits of increasing depth diminish after six layers (hence, why we mentioned **over-smoothing**).
>
> ### QM9 and Kinetic Modeling Observations
>
> In the QM9 experiments, we noted that when computing QM9 labels—such as the HOMO-LUMO gap—different quantum chemical methods produce ev-scale differences (Page 19, Figure 5). Therefore, we should not expect further improvements from machine learning models if they already reach the data uncertainty limit.
>
> In contrast, in the kinetic modeling task (Page 5, Table 1), we found that the "benefit" of scaling in the no-mixer setup cannot compare to the improvement from adding a few layers of MLP or Transformer, which allows direct modeling of interdependence among structural units.

---

> ### Author Response · Authors · 2024-11-20
> **Response to Reviewer 5LXW Part 2**
>
> ## Reply to the paper representation
>
> We sincerely appreciate the reviewer 5LXW’s opinions on the paper structure, we restructured the introduction and the abstract of the paper to emphasize our RQ.
>
> ## Reply to the Novelty Issues
>
> We appreciate reviewer **5LXW** for their thoughtful and constructive feedback and the opportunity to clarify the novelty and contributions of our work.
>
> ### On the Novelty of Our Study
>
> While previous studies, such as Zaidi et al. (2022) [1], have explored the benefits of pre-training in GNNs, our work differentiates itself in several significant ways:
>
> #### **Comprehensive Study of Scaling Behaviors in Geometric GNNs**
> 1. **First of Its Kind:**
>    To our knowledge, this paper presents the first extensive examination of scaling laws specifically for all-atom geometric GNNs in molecular learning and the first study on how pre-trained all-atom graph embeddings can be transferred to protein modeling tasks.
>
> 2. **Wide Range of Configurations:**
>    We investigate not only model size but also aspect ratios, radial cutoffs, architectural choices, and pre-training datasets, providing practical insights into the design and configuration of geometric GNNs.
>
> 3. **Diverse Applications:**
>    Our analysis spans multiple tasks, including kinetic modeling, protein folding classification, force field predictions, and quantum chemical property predictions, showcasing the versatility and limitations of scaling in different contexts.
>
> #### **Insights into Transferability and Expressiveness**
> We delve into how pre-trained all-atom graph embeddings transfer across various downstream tasks, studying the role of embedding dimensionality and the incorporation of token mixing modules (e.g., MLPs, Transformers, GNNs) for protein modeling. Our findings reveal:
> - Simply increasing embedding dimensionality yields diminishing returns.
> - Architectures that model interdependencies among structural units lead to significant performance gains.
>
> #### **Novel Perspectives and Benchmarks**
> 1. **Improved Evaluation Strategies:**
>    By employing scaffold splitting in QM9 and utilizing non-equilibrium datasets like xxMD, we challenge conventional evaluation methods, providing more rigorous assessments of model generalization capabilities.
>
> 2. **Analysis of Scaling Benefits:**
>    We demonstrate that scaling up model parameters does not uniformly improve performance, particularly when data uncertainties impose fundamental limits—a nuance that prior work has not thoroughly explored.
>
> ---
>
> ### On Formal Connections to GNN Properties (Oversmoothing, Underreaching)
>
> We acknowledge that we did not explicitly investigate formal properties like oversmoothing or underreaching, as these topics are not central to our study. Instead, our focus lies on:
> - Empirical scaling behaviors.
> - Transferability of pre-trained embeddings.
> - Practical implications for molecular and protein modeling tasks.
>
> We believe that understanding these empirical trends is a critical step before delving into formal theoretical analyses, providing a foundation for future investigations into these topics.
>
> [1] Zaidi, Sheheryar, Michael Schaarschmidt, James Martens, Hyunjik Kim, Yee Whye Teh, Alvaro Sanchez-Gonzalez, Peter Battaglia, Razvan Pascanu, and Jonathan Godwin. "Pre-training via denoising for molecular property prediction." arXiv preprint arXiv:2206.00133, 2022

---

> ### Author Response · Authors · 2024-11-20
> **Response to Reviewer 5LXW Part 3**
>
> ## Reply to Experiments
>
> ### Regarding the Comparison of ‘Molecular Dynamics’
>
> First of all, we are unsure about what ‘molecular dynamics’ the reviewer **5LXW** is referring to. Timewarp [1] is an enhanced sampling technique that proposes larger MD timesteps to accelerate the simulation, which is not directly comparable to the kinetic modeling task (if this is what the reviewer **5LXW** refers to) included in this paper. Could the reviewer clarify what we are trying to compare here?
>
> ---
>
> ### Regarding the Splitting and Comparison with More Models
>
> As we have explained in the paper, randomly splitting the data without considering generalizability is not meaningful. Scaffold splitting is prevalent in benchmarks such as MoleculeNet. Furthermore, the xxMD paper [2] contains the results of DimeNet++, MACE, etc.
>
> ---
>
> ### Regarding the Lack of Extensive Comparison
>
> We carefully examined the performance and scaling behaviors of different pre-trained GNNs across diverse tasks:
> - **Conformational variety of single molecules** (force field regression, kinetic modeling).
> - **Chemical variety of many molecules** (quantum chemical property prediction).
> - **Conformational variety of peptides and proteins** (kinetic modeling).
> - **Biological variety of many proteins** (folding classification).
>
> In detail, we provided more than 100 data points of pre-training experiments covering two GNN architectures with variations in model depth, width, aspect ratio, and radius cutoff configurations on both equilibrium and non-equilibrium molecular datasets:
> - **xxMD Experiments:** We tested two GNNs (non-pre-trained and pre-trained on PCQM and Denali, respectively) across hidden dimensions on four molecular systems.
> - **QM9 Experiments:** We tested two GNNs (non-pre-trained and pre-trained on PCQM) across hidden dimensions on five tasks.
> - **VAMPNet Experiments:** We evaluated three systems (molecules to proteins) with different embedding dimensions and token mixers.
> - **Folding Classification Task:** We tested ProNet with and without pre-trained all-atom embeddings across different dimensions, representing the most comprehensive and detailed study among relevant works.
>
> ---
>
> ### Regarding the xxMD-Temporal Benchmarks
>
> In the caption of **Table 2 (Page 8)**, we indicated that we compared models pre-trained (with prefix PT) and not pre-trained across different dimensions and targets. In **Table 3 (Page 9)**, we noted that only models pre-trained on the Denali dataset (PT-Denali) could positively transfer. ViSNet, without being pre-trained, performs better on xxMD-Azobenzene and dithiophene subsets.
>
> To better support our claims, we have added an additional table in the appendix with the complete results. We summarized the ViSNet/ET results on azobenzene, stilbene, malonaldehyde, and dithiophene across 64, 128, 256, and 384 dimensions with no pre-training, pre-trained on PCQM, and pre-trained on Denali setups in **[Appendix L, Page 31, Table 9]**.
>
> ## Reply to minor points
> As we have explained in the paper (Page 5 Line 237-240) and the main figure (Page 2 Figure 1), *In each window, atomic structures are treated as individual graphs and processed by the pre-trained Geom-GNN to extract atomic-level features, which are aggregated into residue-level representations or “tokens.” The architecture can employ self-attention (SA), multi-layer perceptron (MLP), or message-passing mechanisms to enhance representational power.*
>
> ## Reply to zero-shot definition
> Using masked language modeling as an example, language models trained to complete the sentence in English can transfer to complete the sentence in Chinese. (Same task, different data domain) In our setup, the backbone network is trained on small molecules with roughly 10-20 heavy atoms with denoising objective, and the backbone network is transferred to infer the atomistic embedding of peptides and proteins, and then a separate head is trained with a different objective. (Different task, different data domain) Since the backbone network has never seen protein systems, we think it’s appropriate to claim it as zero-shot transfer for embedding inference.
>
> ## Reply to pre-training vs. fine-tuning
> Pre-training datasets are broad and diverse, aiming to build general representations, while downstram datasets are task-specific and focused, helping the model adapt to particular applications.
>
>  [1] Klein, Leon, Andrew Foong, Tor Fjelde, Bruno Mlodozeniec, Marc Brockschmidt, Sebastian Nowozin, Frank Noé, and Ryota Tomioka. "Timewarp: Transferable acceleration of molecular dynamics by learning time-coarsened dynamics." Advances in Neural Information Processing Systems (NeurIPS), vol. 36, 2024.
>  [2] Pengmei, Zihan, Liu, Junyu, and Shu, Yinan. "Beyond MD17: the reactive xxMD dataset." Scientific Data, vol. 11, no. 1, 2024, p. 222. Nature Publishing Group UK London.

---

> > ### Comment · Reviewer_5LXW · 2024-11-22
> >
> > I didn't have very much time yet to look thoroughly through the responses and plan to do that later.
> >
> > **Regarding the Comparison of ‘Molecular Dynamics’**
> >
> > It's clear that it is not comparable and might have been also a misunderstanding from my side. But why to choose the kinetic modeling task and not to check how good the pretraining strategy works, e.g., with respect to enhanced sampling techniques for speeding up molecular dynamics?
> >
> > The problem for me with the kinetic modeling task: Are there any publications out on exactly this dataset? Or did you create the dataset yourself? As a reviewer, I would prefer to see independent publications having reported on one and the same dataset/benchmark to be better able to judge how much improvement over previous state-of-the-art there is now.
> >
> > Would it be possible that you try out your method on a task, where already several other independent researchers reported results or point me to publications, that reported results on exactly the kinetic dataset you looked at?
> >
> > **Regarding the Lack of Extensive Comparison**
> >
> > Having investigated pretraining behavior on only two arbitrary (?) GNN architectures in general seems not very convincing to make general statements about pretraining, which should possibly be relevant to a larger field. Moreover one of the two architectures seems to be an extension of the other one as the authors write. Why did the authors not consider architectures like EGNN, SEGNN, PaiNN, Equiformer, NequIP, Allegro, MACE etc.?
> > Why have the authors exactly chosen the two architectures they have chosen? I wonder, what the argumentation is why to compare two related architectures and to ignore others. In case the authors think that experiments for VisNet seem to be exceptionally important, they should possibly also restrict the validitiy of any statement in the title, the abstract and the paper towards this architecture or towards VisNet and ET only,  and not refer to the more general term of "Geom-GNNs".

---

> > > ### Author Response · Authors · 2024-11-22
> > > **Response to Reviewer 5LXW Reply**
> > >
> > > ### Reply to the Comparison of ‘Molecular Dynamics’
> > >
> > > We chose the kinetic modeling tasks for several key reasons:
> > >
> > > 1. **Transferability Assessment:** These tasks allowed us to examine the transferability of pre-trained GNNs on molecule/protein conformational diversity that lies entirely outside the pre-training domain.
> > > 2. **Relevance to the Molecular Dynamics Community:** Kinetic modeling and collective variable (CV) learning have a long history and remain areas of broad interest within the molecular dynamics community.
> > >
> > > We selected the alanine dipeptide and pentapeptide datasets from the publicly available MDShare website. Alanine dipeptide is a simple toy system, where the two backbone dihedrals serve as well-established CVs. Pentapeptides, on the other hand, are widely studied in the relevant literatures. This dataset is used as a beginner’s tutorial in tools like PyEMMA and DeepTime, as introduced by Noé et al. Indeed, we designed **Figure 3 (Page 6)** to be analogous to the PyEMMA tutorial, as noted in the footnote on **Page 5**, allowing the reviewer to directly compare our results with those in that tutorial.
> > >
> > > We also chose the lambda6-85 dataset from Folding@home because it is publicly available, enabling us to illustrate the transferability of pre-trained embeddings to much larger systems. Furthermore, the original paper on lambda6-85 proposed specific CVs that we compared in **Figure 16 (Page 26)**, which itself contains non-trivial efforts. While we acknowledge that long trajectories of fast-folding proteins from D.E. Shaw [1] are commonly used for benchmarking, these datasets are not publicly available, which does not comply with the ICLR guidelines.
> > >
> > > We kindly ask reviewer 5LXW to clarify the meaning of comparing kinetic modeling "with respect to the speeding up of molecular dynamics" or "how effective the pre-training is." If the reviewer is referring to how the learned CVs could be used for biased dynamics, we refer them to related literature such as [2][3]. If the reviewer is instead referring to how effective pre-training is for feature extraction, we suggest consulting **Figure 17 (Page 27)**, which inspects the pre-trained feature space, in conjunction with all the results presented in the main text.
> > >
> > > ---
> > >
> > > ### On the Choice of ViSNet
> > >
> > > As explained earlier, we selected ViSNet due to its outstanding performance and efficiency. As indicated by Wang et al. [4] in the original ViSNet paper, it outperforms all the models mentioned by reviewer 5LXW on numerous datasets, including molecular property prediction and force field prediction, and does so with significantly lower computational costs. This is especially evident when compared to group-equivariant models such as Equiformer, NequIP, Allegro, and MACE.
> > >
> > > Moreover, ViSNet has been carefully benchmarked on a real protein system, **chignolin**, where it significantly outperforms group-equivariant networks like MACE across various conformational states. It is unclear how additional comparisons would benefit practitioners looking to apply all-atom 3D GNNs to protein modeling.
> > >
> > > At the same time, ET (Equivariant Transformer), which is shipped with the TorchMD package [5], has been widely applied in protein dynamics modeling. However, there is a notable lack of similarly comprehensive tests for the other architectures mentioned by the reviewer. We believe our comparisons already provide relevant insights into the effectiveness of ViSNet for practical protein modeling tasks.
> > >
> > > **References**
> > > 1. Lindorff-Larsen, K., Piana, S., Dror, R. O., & Shaw, D. E. (2011). How fast-folding proteins fold. *Science*, 334(6055), 517–520.
> > > 2. Bonati, L., Piccini, G. M., & Parrinello, M. (2021). Deep learning the slow modes for rare events sampling. *Proceedings of the National Academy of Sciences*, 118(44), e2113533118.
> > > 3. Zou, Z., Wang, D., & Tiwary, P. (2024). Graph Neural Network-State Predictive Information Bottleneck (GNN-SPIB) approach for learning molecular thermodynamics and kinetics. *arXiv preprint arXiv:2409.11843*.
> > > 4. Wang, Y., Wang, T., Li, S., He, X., Li, M., Wang, Z., Zheng, N., Shao, B., & Liu, T.-Y. (2024). Enhancing geometric representations for molecules with equivariant vector-scalar interactive message passing. *Nature Communications*, 15(1), 313.
> > > 5. Pelaez, R. P., Simeon, G., Galvelis, R., Mirarchi, A., Eastman, P., Doerr, S., Thölke, P., Markland, T. E., & De Fabritiis, G. (2024). TorchMD-Net 2.0: Fast Neural Network Potentials for Molecular Simulations. *Journal of Chemical Theory and Computation*.

---

> > > ### Author Response · Authors · 2024-11-24
> > > **Further Discussion**
> > >
> > > Dear Reviewer 5LXW,
> > >
> > > We hope our previous response has addressed your concerns. If there are remaining issues, we would be happy to discuss them further. We kindly ask if you could specify which points require clarification, ideally with more detailed setups or comparison to existing literatures. This would greatly assist us in refining our work constructively and addressing your feedback effectively.
> > >
> > > Kind Regards,
> > > Authors

---

> > > ### Author Response · Authors · 2024-11-26
> > > **Kind Reminder Regarding Reviewer Comments**
> > >
> > > Dear **Reviewer 5LXW**,
> > >
> > > As the revision process is nearing completion, we would like to kindly remind you to review our responses to your comments, which addressed your concerns point by point. We hope our explanations have clarified any remaining gaps, as we have already addressed all other reviewers' comments and suggestions.
> > >
> > > If there are any additional points to discuss, we would be happy to engage further, although we regret that we are unable to conduct additional experiments at this stage.
> > >
> > > Thank you for your time and understanding.
> > >
> > > Best regards,
> > > The Authors

---

> > > ### Comment · Area_Chair_8tuF · 2024-11-26
> > >
> > > Hi Reviewer 5LXW,
> > >
> > > Previously, you mentioned, "I didn't have very much time yet to look thoroughly through the responses and plan to do that later." Now the discussion stage is ending, and I hope you can take some time to acknowledge the authors' response.
> > >
> > > At your earliest convenience, could you check the response from the authors, and then indicate if it changes your opinions or not?
> > >
> > > Best,
> > >
> > > AC

---

> ### Comment · Reviewer_5LXW · 2024-11-28
>
> Sorry, it has been a busy time for me. I have read the paper again in light of the review comments and responses to them.
> Overall the author's research goals are more clear to me now. Especially it was not my intention "to frame their work as a continuation of pre-training methods or graph scaling laws, despite our repeated clarifications", but might have misunderstood it.
>
> > We kindly ask reviewer 5LXW to clarify the meaning of comparing kinetic modeling "with respect to the speeding up of molecular dynamics" or "how effective the pre-training is."
>
> Here the authors must have misunderstood me completely with what I meant; However let's not dwell on it further.
>
>
>
> with respect to the dataset for kinetic modeling tasks:
> I find it strange when one has to compare with a tutorial example. It raises concerns, whether this is far from real-world.
>
> with respect to Random Split QM9:
> Here the authors must have misunderstood me. I even marked it as "Strengths", that they considered not only random splitting.
> However, even if results are overoptimistic, many researchers evaluated their methods for this benchmark (which does not mean, that I think it's necessarily a good dataset or benchmark, but), which could nevertheless give additional insights.
>
> I must acknowledge, however, that I am not an expert in each of the specific subfields from which the authors have drawn their datasets. Apparently, other reviewers found the experiments on the selected datasets to be scientifically more valuable than I did (or at least they did not question it). That is one reason why I am considering possibly raising the score.
>
> clarity of the contributions:
> - To me it seems that Geom-GNN is the same as shown in Figure 1 and refers to a specific network architecture. Is this true?
> - Is Geom-GNN a new architecture, a generic term, or, an established architecture from somewhere else?
>   - If it is a new architecture, the authors should declare it as a contribution.
>   - If it is a generic term, then the authors should throughout the paper be more specific which network architecture they are actually referring to.
>   - If it is an established architecture from somewhere else, they should properly cite it.
>
> There are further questions on hyperparameter selection:
> - When an architecture with pretrained features is compared to an architecture with non-pretrained features, is the hyperparameter selection individual for each of the two cases? Maybe architectures with non-pretrained features need more layers. What is the authors' opinion on this?
>
> Question on Scaffold-QM9:
> Why are only 5 columns shown in Table 2? Does Scaffold-QM9 have less targets than QM9? If it does not have less targets, why did you skip the other targets?
>
> presentation of the paper:
> - There is a typo in the first sentence of the paper:
> In silico molecular computation and simulation are indispensable tool**s** in modern research for biology, chemistry, and material sciences capable of accelerating the discovery of new drugs and materials, as well as prediction of protein structures and functions
> - This sentence seem weird to me:
> As we have studied a few preliminary application of pre-trained graph embedding, we wonder if the power of features given by Geom-GNNs with varying architectures and configurations.
> - The paper is hard to read as many figures, etc. are in the appendix and it is not clear with the figure numbers, whether the figure is expected to be found in the main text or in the appendix

---

> > ### Author Response · Authors · 2024-11-28
> >
> > ## Regarding datasets
> >
> > We do not want to judge whether one particular dataset is ‘far from’ real-world or not. However, we could provide those references [1][2][3] here again, as we did in the paper. We also would like to mention that both PyEMMA [4] and Deeptime [5] are widely used packages in the molecular dynamics community. We will add the corresponding citations in the revised version as well.
> >
> > ## Regarding QM9 dataset
> >
> > While the reviewer acknowledged that random-split QM9 is not a good benchmark, there is no benefit in performing an additional set of experiments with 5 (tasks) × 5 (dimensions) × 2 (models) × 2 (pretrained or not), especially since this is not the central focus of the paper. Moreover, benchmarking on an improperly handled dataset, such as random-split QM9, could lead to false conclusions, e.g., the existence of power-law scaling for those tasks. We chose the first five targets to investigate the effect of pre-training and scaling, as five targets are already sufficient to observe the pattern and support our claims, especially considering the wide scope and configurations explored in the paper.
> >
> > ## Regarding Geom-GNNs
> > From lines 033–039, we indicated that Geometric Graph Neural Networks (Geom-GNNs) refer to a class of GNNs that operate with coordinates. From lines 733–741, we briefly extend this discussion. Furthermore, in both the figure and caption of **Figure 1**, we clearly showed a framework where pre-trained Geom-GNNs act as local geometric descriptors to featurize residue-level conformations. We believe the reviewer has already noticed that we chose two architectures, ViSNet [6] and ET [7], as these were properly cited in the paper.
> >
> > It is also a common practice to use a smaller learning rate for pre-trained weights than for randomly initialized weights. Furthermore, it is difficult to see the point of comparing an apple to a pear.
> >
> > ## Regarding formats
> >
> > We thank the reviewer for the formatting suggestions. We also clearly indicated in the paper which figures are expected to be found in the appendix as **Figure X (Appendix)**. Moreover, we chose to represent those results as figures because either they cannot be effectively represented as tables or they are easier to interpret as figures.
> >
> >
> > ### References
> >
> > 1. Mardt, A., Pasquali, L., Wu, H., et al. *VAMPnets for deep learning of molecular kinetics.* Nature Communications, 9, 5 (2018). [https://doi.org/10.1038/s41467-017-02388-1](https://doi.org/10.1038/s41467-017-02388-1)
> >
> > 2. Nüske, F., Wu, H., Prinz, J.-H., Wehmeyer, C., Clementi, C., & Noé, F. *Markov state models from short non-equilibrium simulations—Analysis and correction of estimation bias.* Journal of Chemical Physics, 146(9), 094104 (2017). [https://doi.org/10.1063/1.4976518](https://doi.org/10.1063/1.4976518)
> >
> > 3. Bowman, G. R., Voelz, V. A., & Pande, V. S. *Atomistic folding simulations of the five-helix bundle protein λ6−85.* Journal of the American Chemical Society, 133(4), 664-667 (2011). [https://doi.org/10.1021/ja106844r](https://doi.org/10.1021/ja106844r)
> >
> > 4. Scherer, M. K., Trendelkamp-Schroer, B., Paul, F., Pérez-Hernández, G., Hoffmann, M., Plattner, N., ... & Noé, F. *PyEMMA 2: A software package for estimation, validation, and analysis of Markov models.* Journal of Chemical Theory and Computation, 11(11), 5525-5542 (2015). [https://doi.org/10.1021/acs.jctc.5b00743](https://doi.org/10.1021/acs.jctc.5b00743)
> >
> > 5. Hoffmann, M., Scherer, M., Hempel, T., Mardt, A., de Silva, B., Husic, B. E., ... & Noé, F. *Deeptime: A Python library for machine learning dynamical models from time series data.* Machine Learning: Science and Technology, 3(1), 015009 (2021). [https://doi.org/10.1088/2632-2153/ac2a50](https://doi.org/10.1088/2632-2153/ac2a50)
> >
> > 6. Wang, Y., Wang, T., Li, S., et al. *Enhancing geometric representations for molecules with equivariant vector-scalar interactive message passing.* Nature Communications, 15, 313 (2024). [https://doi.org/10.1038/s41467-023-43720-2](https://doi.org/10.1038/s41467-023-43720-2)
> >
> > 7. Thölke, P., & De Fabritiis, G. (2022). *Torchmd-net: Equivariant transformers for neural network-based molecular potentials.* arXiv preprint arXiv:2202.02541. [https://arxiv.org/abs/2202.02541](https://arxiv.org/abs/2202.02541)

---

> > > ### Comment · Reviewer_5LXW · 2024-11-29
> > >
> > > > From lines 033–039, we indicated that Geometric Graph Neural Networks (Geom-GNNs) refer to a class of GNNs that operate with coordinates. From lines 733–741, we briefly extend this discussion. Furthermore, in both the figure and caption of Figure 1, we clearly showed a framework where pre-trained Geom-GNNs act as local geometric descriptors to featurize residue-level conformations. We believe the reviewer has already noticed that we chose two architectures, ViSNet [6] and ET [7], as these were properly cited in the paper.
> > >
> > > If the architecture in Figure 1 is an own contribution from the authors, I really recommend to give it a name and restrict considerations and conclusions to this architecture. There might be other options than the architecture in Figure 1, how to make use of pretrained features, and such other architectures could show a much different scaling behavior.
> > >
> > > Do the authors think that they are the first architecture at all, which makes use of pretrained geometric features?
> > >
> > >
> > > > Furthermore, it is difficult to see the point of comparing an apple to a pear.
> > >
> > > I don't get the point on apples and pears. Is it an answer to my question on hyperparameter selection?
> > > It is the task of the authors to provide clear explanations and not hide what they did. It is important to know this for replicability and for judging and understanding their statements on scaling behaviors etc.\
> > > If hyperparameter selection has been done differently for different types of pretrained features, pretrained and non-pretrained features, different datasets, this has to be described thoroughly.\
> > > If it has not been done individually, it has also to be described and is possibly even more important to know.
> > >
> > > > We also clearly indicated in the paper which figures are expected to be found in the appendix as Figure X (Appendix).
> > >
> > > This is not true. The authors should for example look at their sentence "Per Figure 20, we observe improvement of integrating
> > > such all-atom features into the ProNet without angle information." Figure 20 is somewhere in the appendix, although it is not declared in this sentence to be in the appendix.

---

> > > > ### Author Response · Authors · 2024-11-29
> > > >
> > > > We believe it is very important to understand the representational power of the underlying 3D GNN models; thus, we thoroughly studied the self-supervised pre-training and supervised fine-tuning in the atom-scale tasks to estimate the resulted features. We also thank the reviewer for acknowledging all the novel results presented in the paper, where those pre-trained features can be generally applicable to a variety of downstream tasks such as embedding/conformational analysis, kinetic modeling, enhancing existing architectures as stronger features than conventional descriptors, and we thoroughly studied the scaling behaviors at different levels.
> > > >
> > > > Regarding the reviewer’s suggestion of adding an additional comparison of pre-trained models (apple) with non-pre-trained models with a different number of layers (pear), we believe this would not be a fair comparison as the variables are not controlled. Additionally, we are not hiding any details about hyperparameter selection, and we kindly refer the reviewer to **Appendix F** for all the hyperparameters.
> > > >
> > > > We thank the reviewer for catching the formatting issues. We will thoroughly ensure that all the figures in the appendix are annotated as **Figure X (Appendix)** in the revised version.

---

> ### Comment · Reviewer_5LXW · 2024-12-02
>
> > Regarding the reviewer’s suggestion of adding an additional comparison of pre-trained models (apple) with non-pre-trained models with a different number of layers (pear), we believe this would not be a fair comparison as the variables are not controlled. Additionally, we are not hiding any details about hyperparameter selection, and we kindly refer the reviewer to Appendix F for all the hyperparameters.
>
> I am not suggesting new comparisons.
> But, I wonder, whether it **has to be declared** as a limitation of your work.
>
> 1.
>
> In case you are purely doing a comparison of a fixed network architecture and other hyperparameters (up to those network architecture design parameters and  hyperparameters of study) once with pre-trained features and once without and study, e.g. the effect of model width, it seems ok to my understanding.
>
> 2.
>
> However, I wonder, e.g., for Table 3. In case you are doing a general comparison in the way like "Do pretrained features have advantages over non-pretrained features", then one would expect an individual hyperparameter optimization for both of these options. I am not absolutely sure, what the goal with Table 3 exactly was, though.
>
> to be more clear, what I mean, I formulate it as an exemplary question to you: Consider you would need to give advice to a machine learning engineer, whose task is to train a model with a minimum validation error. The engineer can either take pretrained features or non-pretrained features. Would you recommend the engineer to a) optimize hyperparameters once for pretrained and once for non-pretrained features or would you recommend b) that the best hyperparameters found with one type of the features also have to be used for the other type of features?
>
> If you opt for a) then you should possibly declare lack of individual hyperparameter optimization as a limitation of your work, in case your goal is/was to show that one type of features is better than the other one. Otherwise it could be misleading to my opinion.

---

> > ### Author Response · Authors · 2024-12-02
> >
> > We thank the reviewer for clarifying their new questions.
> >
> > In **Table 3**, together with **Table 9 (Appendix)**, we reported the base GNN models (pre-trained or not pre-trained) on xxMD temporal split. From **Line 440–451**, we explained the results showing that only models pre-trained with non-equilibrium structures exhibit positive transfer. Another phenomenon we discussed in the text is that ET is a less-performing model compared to ViSNet. However, ET demonstrates positive transfer on 4/4 tasks, while ViSNet only shows positive transfer on 2/4 tasks, even though ViSNet outperforms ET in all tasks. In **Lines 450–451**, we further clarified that this observation can be attributed to the DFT data uncertainty. We do not aim to grid search for a set of hyperparameters to minimize the test error of a specific dataset as this is beyond the scope of the paper. But thanks for your suggestion and we will add a clarification sentence in conclusion.
> >
> > We appreciate the reviewer for the amazing feedbacks to help improved the quality of the manuscript. As we have provided additional data and experimental results, and responded to all the comments, we hope the reviewer can change their score.

---

> ### Comment · Reviewer_5LXW · 2024-12-03
>
> I am not super-convinced the paper should be accepted;
> Due to the discussion with the authors their research goal became more clear to me. I have the impression that the relevance of this work is a little bit limited. The empirical investigations might have a clear novel aspect, although to me unclear how interesting they are to the community.
>
> At the same time, my point of view might be subjective and I have to recognise that others see the manuscript much more positive.
> Since I am not the person, who wants to block a potentially useful publication, I raise my score to 6.
>
> for others who read my one or two initial reviews:
> There might have been misunderstandings from my side. I do not change the text (and the initial ratings except the overall rating) at this stage any more, but please do not necessarily rely/refer to it.

---

> > ### Author Response · Authors · 2024-12-03
> >
> > Thank you very much for your insights and time engaging the discussion, and the recognition of the novelty regarding our empirical investigations.

---

### Official Review · Reviewer_i3nd · 2024-11-02

**Soundness:** 3
**Presentation:** 4
**Contribution:** 3
**Rating:** 8
**Confidence:** 3

**Summary:**

This work presents a novel investigation into whether pre-trained Geom-GNNs (Graph Neural Networks for Conformational Molecules) possess efficient and transferable geometric representation capabilities, particularly in addressing the low generalization of models typically trained on specific tasks. The authors also aim to introduce “Neural Scaling Laws” to summarize the performance behavior of these pre-trained Geom-GNNs. However, it is unfortunate that the experimental results indicate that Geom-GNNs do not adhere to power-law scaling laws and fail to demonstrate predictable scaling behavior across various supervised tasks. Furthermore, the findings reveal that the all-atom embedding graph representations derived from Geom-GNNs exhibit excellent expressive capabilities, suggesting that Geom-GNNs can function effectively as zero-shot transfer learners.

**Strengths:**

1. **Innovation**: The study presents a novel perspective, exploring an area that remains under-researched (to my knowledge), offering significant insights for the development of Geom-GNNs.
2. **Clarity and Structure**: The article is well-organized, with clear presentation and summary of experiments and viewpoints, facilitating reader comprehension.
3. **Robust Experimentation**: The experimental design is thorough, effectively supporting the authors’ conclusions.
4. **Exploration of Zero-Shot Transfer Capability**: The investigation into the zero-shot transfer ability of Geom-GNNs is intriguing, with experiments indicating their potential as excellent zero-shot transfer learners.
5. **Pre-training Insights**: Through extensive denoising pre-training tasks, valuable experiences have been gained regarding the pre-training of Geom-GNNs, including aspects such as model width, depth, aspect ratio, and the cutoff radius in geometric atomic graph construction, providing rich guidance for pre-training.
6. **Advancement of Unified Architecture**: Given the widespread attention and efforts in the research of all-atom Geom-GNNs, this study effectively inspires researchers to reconsider the design of Geom-GNN architectures and the adjustment of training strategies, thereby promoting the development of a unified Geom-GNN architecture.

**Weaknesses:**

1. In Figure 6, which explores different model widths, even though the x-axis represents the total number of parameters (with model depth held constant), it would be more beneficial to indicate the model width for each point in the legend to enhance result presentation. Similarly, in Figures 4 and 7, which demonstrate the impact of model depth, using the legend to specify the exact number of layers might be more effective. In general, clear legends are always advantageous.
2. The comparison between models trained from scratch and those fine-tuned in Section 6.1 could be more comprehensive if extended to include model depth. Previous discussions (albeit in the context of pre-training) have presented certain viewpoints, and it is anticipated that these would also have significant effects during fine-tuning.

**Questions:**

as Weaknesses.

---

> ### Author Response · Authors · 2024-11-20
> **Response to Reviewer i3nd**
>
> ## Summary
>
> We sincerely appreciate the reviewer **i3nd** for the careful proofreading of our paper, which have helped improve the quality of the manuscript. our paper aims to answer a previously unanswered research question in graph representational learning: **"Are pre-trained all-atom geometric graph neural network (GNN) representations transferable to protein modeling, and how expressive are they?"**
>
> To answer this research question, our contributions are as follows:
> 1. **Scaling Behaviors of Geometric GNNs:**
>    We studied the scaling behaviors of state-of-the-art geometric GNNs in unsupervised, self-supervised, and supervised setups, rather than focusing on creating new architectures or pre-training objectives.
> 2. **Demonstrating Transferability:**
>    We pre-trained these GNNs on small molecular datasets and demonstrated their transferability to proteins with all-atom resolution, highlighting their expressiveness in these settings.
>
> ### Regarding the figure formatting
>
> We took the reviewer's advice and revised the figures accordingly in the manuscript for clearer representation. Per other reviewers’ suggestions, we also restructured the introduction and abstract sections to highlight our RQ.
>
> ### Regarding the comparison of model depth
>
>
> We appreciate the reviewer for suggesting comparing the effect of model aspect ratio in the fine-tuning stage. We did the following extra experiments on QM9 to illustrate the effect by fixing the parameter count and varying the depth of the model (this table is integrated in Appendix L as Table 10 on Page 32):
>
> | **Layer-Dim** | **Setup**  | **ε_LUMO** | **Δϵ** |
> |---------------|------------|------------|--------|
> | **6L-128**    | No PT      | 42.8       | 94.5   |
> |               | PT on PCQM | 35.9       | 80.7   |
> | **5L-144**    | No PT      | 46.1       | 104.8  |
> |               | PT on PCQM | 39.5       | 88.3   |
> | **4L-160**    | No PT      | 45.6       | 98.5   |
> |               | PT on PCQM | 39.8       | 96.7   |
> | **3L-184**    | No PT      | 44.5       | 98.4   |
> |               | PT on PCQM | 44.1       | 103.0  |
> | **2L-216**    | No PT      | 43.5       | 97.6   |
> |               | PT on PCQM | 46.6       | 96.0   |
>
> Interestingly, we found there is no obvious trend of aspect ratio effect on the results of blank models, but there is a trend for pre-trained models where deeper models perform better than shallower models. We would also like to refer the reviewer to Figure 2 in Li et al. 2024 [1], where deeper models generally perform better than shallower models on force field tasks.
>
> [1] Li, Yunyang, Yusong Wang, Lin Huang, Han Yang, Xinran Wei, Jia Zhang, Tong Wang, Zun Wang, Bin Shao, and Tie-Yan Liu. "Long-short-range message-passing: A physics-informed framework to capture non-local interaction for scalable molecular dynamics simulation." arXiv preprint arXiv:2304.13542, 2023.

---

> > ### Comment · Reviewer_i3nd · 2024-11-23
> >
> > Thank you to the authors for their hard work and thoughtful response in addressing the identified weaknesses. I will maintain my score as it stands, as I am not an expert in this field, and the initial score I provided is already sufficiently high from my perspective. Thank you again!

---

> > > ### Author Response · Authors · 2024-11-23
> > > **Thank you**
> > >
> > > We really appreciate your helpful reviews and recognition of the novelty. We wish you a good break :)
> > >
> > > Kind Regards,
> > > Authors

---

### Official Review · Reviewer_KcDu · 2024-11-02

**Soundness:** 3
**Presentation:** 3
**Contribution:** 2
**Rating:** 6
**Confidence:** 4

**Summary:**

The paper investigates denoising pretraining and potential scaling laws for geometric graph neural networks (GNNs) on supervised tasks. These GNNs are pre-trained on noisy coordinate data to learn how to denoise and reconstruct the original coordinates. The effectiveness of this approach is tested on various downstream applications, including molecular kinetics, fold classification, and energy and force prediction. Additionally, the paper examines the scaling behavior of these models and highlights specific limitations in supervised prediction tasks.

**Strengths:**

1. Demonstrates substantial performance improvements in kinetic modeling on VAMP score metrics by utilizing denoising pretraining techniques.
2. Applies the self-supervised pretraining approach to a variety of downstream tasks, successfully proving its effectiveness across applications.
3. Examines scaling laws in both standard equivariant models and pre-trained equivariant models, finding that even pre-trained models diverge from typical neural scaling laws due to issues like under-reaching and over-smoothing. The author suggests that, while scaling models has its benefits for supervised and unsupervised tasks, it may be more effective to focus on addressing data label uncertainty and using active token mixing to mitigate information bottlenecks.

**Weaknesses:**

1. The author claims to be the first to demonstrate the pre-trained Geom-GNNs’ capability as zero-shot transfer learners. However, I would consider this more appropriately described as downstream task fine-tuning. In zero-shot learning, a model test on data source is directly applied to unseen data class without additional training (e.g., training on English and French, then testing on Chinese without further adjustments). Unlike this approach for molecular kinetics, the paper involves training a separate network with the VAMP score objective.
2. The pretraining methods in this work are limited to coordinate denoising. Other approaches [1,2] that leverage both node mask prediction and coordinate denoising have already proven effective.
3. Although the paper demonstrates the effectiveness of ET and VisNet on several tasks, it does not include evaluations on invariant feature-based networks (such as SchNet, DimeNet, or GemNet) or tensor product-based networks like Equiformer and MAC.

[1] Cui, Taoyong, et al. "Geometry-enhanced pretraining on interatomic potentials." Nature Machine Intelligence 6.4 (2024): 428-436.
[2] Zhou, Gengmo, et al. "Uni-mol: A universal 3d molecular representation learning framework." (2023).

**Questions:**

1. The paper claims to demonstrate zero-shot transfer learning using pre-trained Geom-GNNs. However, the described method seems closer to downstream task fine-tuning, given that a separate network is trained with the VAMP score objective. Could the authors clarify how this approach qualifies as zero-shot transfer rather than fine-tuning?

2. Limited Pre-Training Approaches:
The pre-training in this work is restricted to coordinate denoising. Given that prior work has successfully used a combination of node mask prediction and coordinate denoising for improved performance. Specifically, how might adding a node mask objective influence for the molecular kinetics tasks? Would it enhance the model’s ability to generalize across different molecular conformations?
Additionally, could the authors hypothesize the potential impact of such an extended pre-training approach on scaling behavior?

3. While the effectiveness of ET and ViSNet is demonstrated on several tasks, the study lacks comparisons with invariant feature-based networks (e.g., SchNet, DimeNet, GemNet) and tensor product-based networks (e.g., Equiformer, MAC). Could the authors provide insights into how their method might perform relative to these alternative architectures?

---

> ### Author Response · Authors · 2024-11-20
> **Response to Reviewer KcDu Part 1**
>
> ## Summary
>
> We appreciate the thoughtful comments and summary from reviewer **KcDu**, which have helped improve the quality of the manuscript. Before proceeding to further discussions, we would like to clarify several key points.
>
> Our paper does not focus on extending denoising pre-training techniques, investigating pre-training task choices, or developing new model architectures. Instead, it addresses a previously unanswered research question in graph representational learning:
> **"Are pre-trained all-atom geometric graph neural network (GNN) representations transferable to protein modeling, and how expressive are they?"**
>
> To answer this research question, our contributions are as follows:
> 1. **Scaling Behaviors of Geometric GNNs:**
>    We studied the scaling behaviors of state-of-the-art geometric GNNs in unsupervised, self-supervised, and supervised setups, rather than focusing on creating new architectures or pre-training objectives.
> 2. **Demonstrating Transferability:**
>    We pre-trained these GNNs on small molecular datasets and demonstrated their transferability to proteins with all-atom resolution, highlighting their expressiveness in these settings.
>
> ---
>
> ### Performance Increase on VAMP and Fold Classification
>
> We believe the observed performance increases on the VAMP objective and fold classification task do not stem from specific pre-training techniques. Instead, the improvements result from the fact that we pre-trained geometric GNNs, then transferred them in a zero-shot fashion and organically combined with higher-level architectures (Page 5 Table 1). Similarly, the reason why the fold classification results (Page 30 Figure 20) got improved originates from the fact that the inferred graph embedding from pre-trained GNNs contain rich all-atom information instead of applying denoising pre-training to downstream tasks. (And we did not)
>
> ### Zero-shot Transfer Learners
> We appreciate the reviewer for noticing the difference of zero-shot learning setup here when comparing to natural language modeling and computer vision. Using masked language modeling as an example, language models trained to complete the sentence in English can transfer to complete the sentence in Chinese. (Same task, different data domain) In our setup, the backbone network is trained on small molecules with roughly 10-20 heavy atoms with denoising objective, and the backbone network is transferred to infer the atomistic embedding of peptides and proteins and then a separate head is trained with a different objective. (Different task, different data domain) Since the backbone network has never seen protein systems, we think it’s appropriate to claim it as zero-shot transfer for embedding inference.
>
> ### About other possible pre-training objectives
> We appreciate the reviewer for suggesting other possible pre-training objectives. Indeed, there are many other pre-training strategies, such as Noisy Node [1], GraphMVP [2], Frad [3], Uni-Mol [4], etc. The scope of our paper does not lay in comparing the effectiveness of various pre-training strategies nor absolute performance. We chose coordinate denoising as our pre-training objective for its proven effectiveness and simplicity, as the coordinate denoising just has the level of additive Gaussian noise as the only hyperparameter. Combining other objectives inevitably requires more complicated hyperparameter scanning. Considering our extensive experiments in both pre-training and downstream tasks, it would be impractical to contain those comparisons in this paper given the computational budget, where including those comparisons could easily multiplicatively increase the workload.
>
> ### Adding a node mask objective
> In kinetic modeling task, we do not think add a node attribute masking pre-training task (e.g. flipping the atom types) would considerably affect the results, since VAMP aims to find the slow modes of the objective movements, as structural information in a single molecular system where all atom types are fixed across samples. Generally, if the two pre-training objectives focus on different perspectives (coordinate denoising and guessing the right atom type), more parameters would be needed to reach the same loss on each pre-training objective.
>
> [1] Godwin et al. "Simple GNN Regularisation for 3D Molecular Property Prediction and Beyond." International Conference on Learning Representations (ICLR), 2022
> [2] Liu et al. "Pre-training molecular graph representation with 3D geometry." arXiv preprint arXiv:2110.07728, 2021.
> [3] Feng et al. "Fractional Denoising for 3D Molecular Pre-training." International Conference on Machine Learning (ICML), 2023, pp. 9938–9961. PMLR.
> [4] Zhou et al. "Uni-mol: A universal 3D molecular representation learning framework." 2023.

---

> ### Author Response · Authors · 2024-11-20
> **Response to Reviewer KcDu Part 2**
>
> ### About including invariant feature-based networks
>
> Let’s reiterate here, we did not focus on a specific GNN architecture nor a specific pre-training technique, and we did not propose a new GNN architecture nor a new pre-training technique in this paper. May the reviewer clarify what we should compare?
>
> Though this discussion is far from the focus of the paper, we can briefly talk about different forms of geometric GNNs. Theoretically, O(d)-equivariant functions can be universally expressed by a collection of scalars [5]. And practically the performance of GemNet/Equiformer/ViSNet/MACE/etc. are not that different on various benchmarks. In line with GVP, PaiNN/ET/ViSNet directly track the vector features by parameterizing the displacement vectors, and those features can be used to predict vector quantities. As the reviewer suggested, the expressiveness of invariant-feature based GNNs increases by the order of features used (SchNet: bond length, DimeNet: bond length/bond angle, GemNet: bond length/bond angle/dihedrals), but higher-order features, for instance, dihedrals require enumerating four indices at the same time. ViSNet includes higher-order features by taking the product of vector representations of source and target nodes, resulting in higher efficiency. As invariant networks do not directly track the vector features, we would have to take the derivative of the scalar output with respect to the input coordinates to predict vector values, where we do not see a point to do so. Regarding group-equivariant networks, we do not foresee considerable differences as explained earlier except for more hyperparameters tuning and computational cost. Again, we would like to emphasize the fact that comparing different GNN architectures or pre-training objectives is not the goal of this paper, and adding this comparison could also easily multiplicatively increase the workload for evaluating every pre-training and downstream task.
>
> [5] Villar, Soledad, David W. Hogg, Kate Storey-Fisher, Weichi Yao, and Ben Blum-Smith. "Scalars are universal: Equivariant machine learning, structured like classical physics." Advances in Neural Information Processing Systems (NeurIPS), edited by A. Beygelzimer, Y. Dauphin, P. Liang, and J. Wortman Vaughan, 2021.

---

> ### Comment · Reviewer_KcDu · 2024-11-22
> **Response to Rebuttal by Authors**
>
> Thank you for your detailed response. However, I still have some concerns based on your explanations.
>
> # Zero-shot Transfer Learners
>
> I agree with your claim only to the extent that it represents "zero-shot transfer for embedding inference." If this is indeed what you mean, I strongly recommend explicitly clarifying this in the paper to prevent confusion among readers. Please highlight that the "zero-shot transfer" applies specifically to embedding inference, as the current phrasing may mislead readers into thinking it applies to the entire model.
>
> However, I maintain that this setup aligns more closely with a standard downstream fine-tuning approach. The model is pre-trained using self-supervised tasks and subsequently fine-tuned with task-specific heads. For example, the kinetic modeling task, as you described, involves further training on the VAMP score. This implies that the entire network(with the backbone parameter frozen) used for the task is trained specifically for this task and has seen task-specific data during fine-tuning. Therefore, using the term "Zero-shot Transfer Learning" throughout the paper without clear qualifications may be misleading. I advise you to avoid such potentially thrilling terminology without sufficient explanation and to be precise about where and how zero-shot transfer applies in your work.
>
>
> # Scaling Behavior
>
> I raised concerns regarding the experimental settings, particularly about why other pretraining objectives and models were not utilized. While I understand that pretraining on the PCQM4M dataset is computationally expensive and it may not be feasible to conduct additional experiments during the review, I believe this limitation affects the paper's broader claims.
> As an experiment-driven paper, such conclusions should be supported by broader experimental settings and benchmarks. It is insufficient to simply discuss how information is processed and aggregated and then claim that models are philosophically similar. Such statements do not substantiate general claims about scaling behaviors.
>
> The experimental settings in this work are limited and do not justify general conclusions. Furthermore, as cited by the authors, existing works [1,2,3] have already explored scaling laws on graphs, testing a range of models to ensure the generalizability of their findings. A similar approach would strengthen the current paper's claims, ensuring that its results are broadly applicable
>
> ## References:
>
> [1] Frey, Nathan C., et al. “Neural scaling of deep chemical models.” Nature Machine Intelligence 5 (2023): 1297–1305.
> [2] Chen, Dingshuo, et al. "Uncovering neural scaling laws in molecular representation learning." Advances in Neural Information Processing Systems 36 (2024).
> [3] Liu, Jingzhe, et al. "Neural scaling laws on graphs." arXiv preprint arXiv:2402.02054 (2024).
>
> # Additional Minor Points
>
> The paper claims to characterize the scaling behaviors of Geometric GNNs in unsupervised setups. However, I could not find any unsupervised tasks presented or analyzed across the paper. This claim is mentioned in the abstract and conclusion but is unsupported in the main text.

---

> > ### Author Response · Authors · 2024-11-22
> > **Response to Reviewer KcDu Reply Part 1**
> >
> > We thank the reviewer KcDu for their suggestion regarding the presentation, and we will clarify in the paper that the supervised, self-supervised, and unsupervised setups pertain solely to the 3D representational models. Similarly, we associate the unsupervised setups with kinetic modeling and fold classification, while the representational model remains fixed during embedding inference.
> >
> > We would like to reiterate that while pre-training on PCQM datasets is feasible, testing the resulting models across the comprehensive evaluations and tasks in our paper is not. We also encourage the reviewers to focus on our research question: **"Are pre-trained all-atom geometric graph neural network (GNN) representations transferable to protein modeling, and how expressive are they?"**
> >
> > ## Reply to Scaling Behavior
> >
> > We respectfully disagree with the reviewer KcDu’s comments regarding our limited experimental setups, the lack of comparison with more pre-training methods and models, and the claim that our results are not generally or broadly applicable. We would like to remind reviewers that the models discussed in [1] and [3] are not pre-trained, and that the models in [2] are pre-trained solely using the GraphMAE objective. Moreover, the literature focuses on specific supervised task types and objects, limited to small molecules, such as molecular property prediction and force field regression. Per our reply to the reviewer **GND7**, our study thoroughly examines the performance and scaling behavior of pre-trained GNNs across diverse tasks, including:
> >
> > - **Single-molecule conformational variety:** Force field regression and kinetic modeling.
> > - **Multi-molecule chemical variety:** Quantum property prediction.
> > - **Peptide/protein conformational variety:** Kinetic modeling.
> > - **Protein biological variety:** Folding classification.
> >
> > We present over 100 pre-training experiments, testing two GNN architectures with varied depth, width, aspect ratio, and radius cutoff configurations on equilibrium and non-equilibrium datasets. Key experiments include:
> >
> > 1. **VAMPNet:** Three systems (molecules to proteins) tested with different embedding dimensions and token mixers.
> > 2. **Folding classification:** ProNet (with/without pre-trained all-atom embeddings) tested across dimensions, representing the most comprehensive study in relevant literature.
> > 3. **Force field prediction:** Two GNNs (non-pre-trained and pre-trained on PCQM and Denali) evaluated across hidden dimensions on 4 molecular systems from xxMD.
> > 4. **Molecular property prediction:** Two GNNs (non-pre-trained and pre-trained on PCQM) evaluated across hidden dimensions on 5 tasks.
> >
> > Since the reviewer KcDu mentioned the literatures [1], [2], and [3], we first provide a brief overview of the key experiments and setups in those works to facilitate a discussion:
> >
> > ### [1]
> > - **Figure 3:** Training budget scaling of non-pre-trained 3D SchNet, PaiNN, and SpookeyNet models on the MD17 datasets with 10,000 randomly split samples, varying batch size and learning rate (force field task). The authors claimed that longer training times yield better performance. However, as criticized by the authors of the xxMD paper, testing models on randomly split MD17 datasets leads to unreliable results due to the strong correlation between train, validation, and test sets.
> > - **Figures 5/A.3/A.4:** Dataset size scaling and parameter scaling of non-pre-trained 3D SchNet, PaiNN, and Allegro models on randomly split ANI1 datasets (force field task). Importantly, model aspect ratio and radius cutoff were not controlled. In contrast, we explored these two aspects in our work and found them to be important hyperparameters.

---

> > ### Author Response · Authors · 2024-11-22
> > **Response to Reviewer KcDu Reply Part 2**
> >
> > ### [2]
> > - **Figure 3:** Dataset size scaling of 2D GIN (pre-trained with GraphMAE using PCQM), SMILES (one-layer Transformer), and Fingerprint (one-layer Transformer) models on three randomly split MoleculeNet subsets (molecular property prediction).
> > - **Figure 4:** Pre-trained versus non-pre-trained 2D GIN on dataset size scaling for three randomly split MoleculeNet subsets (molecular property prediction).
> > - **Figure 5:** Dataset size scaling of 2D GIN on three MoleculeNet subsets with different splits (random, scaffold, and imbalance) (molecular property prediction).
> > - **Figure 6:** Dataset size scaling and parameter scaling of 2D GIN on randomly split MoleculeNet subsets (molecular property prediction). Aspect ratios were not controlled.
> > - **Figure 8:** Dataset size scaling of non-pre-trained 3D PaiNN, SphereNet, and SchNet models on three randomly split QM9 subsets (molecular property prediction).
> > - **Figure 9:** Similar to Figure 5, but using a one-layer Transformer with fingerprints.
> >
> > ### [3]
> > - **Figures 2-3/10-12:** Dataset size scaling and parameter scaling of non-pre-trained 2D GIN and GCN models on PCQM and PPA datasets (molecular property prediction).
> > - **Figure 4:** Parameter scaling of non-pre-trained 2D GIN, GCN, SAT, and GPS models on PPA (molecular property prediction).
> > - **Figures 18/19:** Parameter scaling of non-pre-trained 2D GIN and GCN models on Reddit, HIV, and PCBA datasets (molecular property prediction).
> >
> > ---
> >
> > ### Comparison to Our Work
> > In comparison to Literature [1]—the only 3D GNN scaling study—which focuses on force field tasks with non-pre-trained models in a supervised fashion, we have studied a wider variety of tasks beyond force field tasks, using both pre-trained and non-pre-trained models across supervised, self-supervised, and unsupervised settings. Additionally, we employed statistically more rigorous dataset splits for each task. Furthermore, we proposed the inference of all-atom embeddings in combination with other higher-order architectures.
> >
> > Compared to Literatures [2] and [3], which focus on 2D graph models, we conducted a comprehensive study of 3D GNNs across a variety of setups and tasks, as detailed earlier.
> >
> > ## References:
> > [1] Frey, Nathan C., et al. “Neural scaling of deep chemical models.” Nature Machine Intelligence 5 (2023): 1297–1305. [2] Chen, Dingshuo, et al. "Uncovering neural scaling laws in molecular representation learning." Advances in Neural Information Processing Systems 36 (2024). [3] Liu, Jingzhe, et al. "Neural scaling laws on graphs." arXiv preprint arXiv:2402.02054 (2024).

---

> > ### Author Response · Authors · 2024-11-24
> > **Further Discussion**
> >
> > Dear Reviewer KcDu,
> >
> > We hope our previous response has addressed your concerns. If there are remaining issues, we would be happy to discuss them further. We kindly ask if you could specify which points require clarification, ideally with more detailed setups or comparison to existing literatures. This would greatly assist us in refining our work constructively and addressing your feedback effectively.
> >
> > Kind Regards,
> > Authors

---

> ### Comment · Reviewer_KcDu · 2024-11-25
> **Official Comment by Reviewer KcDu**
>
> Thank you for the detailed response to my comments. I appreciate the clarifications provided regarding the experimental setups, the scope of the study, and the comparisons to related works. The authors’ comprehensive exploration of pre-trained and non-pre-trained 3D GNNs across diverse tasks, coupled with their use of rigorous dataset splits, thoroughly addresses my concerns. Additionally, I acknowledge that the distinctions between the supervised, self-supervised, and unsupervised setups have already been reiterated and clarified in the article. These points, combined with the authors’ thorough responses, address my concern, and I am willing to raise my score.

---

> ### Author Response · Authors · 2024-11-25
> **Thank you**
>
> We appreciate the reviewer for engaging in the discussion and going through all the responses. We wish you a good break :)
>
> Kind Regards,
> Authors

---

### Official Review · Reviewer_GND7 · 2024-11-03

**Soundness:** 2
**Presentation:** 2
**Contribution:** 3
**Rating:** 6
**Confidence:** 4

**Summary:**

In this paper the authors extend previous work on scaling laws for all-atom graph neural networks applied to self-supervised and supervised training. They investigate aspects of pre-training task choice, different downstream evaluations, GNN model size and aspect ratio, as well as the radial cutoff for constructing nearest neighbor graphs.

**Strengths:**

The paper continues an important line of work in exploring the utility of pre-training and scaling GNNs for molecular learning tasks. Unlike other dominant areas of deep learning, all-atom molecular representation learning relies on GNNs and does not directly benefit from advances in scaling sequence-based models.
The authors explore model size, aspect ratio, nearest neighbor cutoff radius, and architecture to provide a comprehensive look into the scaling behavior of molecular GNNs.

**Weaknesses:**

Like most molecular GNN works, the authors are limited by the available evaluations (e.g., QM9). xxMD is an interesting evaluation, but these datasets are limited to a small set of specific molecules. QM9 with B3LYP in particular is not informative, and the authors might consider newer benchmarks like POLARIS or a subset of the Therapeutic Data Commons to strengthen their evaluations.
The paper does not offer a clear and concise summary of recommendations based on the empirical findings, which is essential to achieving the stated aim of inspiring practitioners to rethink GNN training.

**Questions:**

Many of the evaluations are on datasets for specific molecules, which are very useful for understanding specific model behavior, but should be complemented by more general evaluations, especially in the context of examining scaling behavior. Can the authors comment on or provide additional evidence that these specific, bespoke evaluations are connected to more general relationships between pre-training setups and downstream evaluations?

---

> ### Author Response · Authors · 2024-11-20
> **Response to Reviewer GND7 Part 1**
>
> ## Summary
>
> We sincerely appreciate reviewer **GND7** for their thoughtful feedback, which has significantly improved the manuscript's quality. We are grateful for their recognition of our contribution and the novelty of using self-supervised geometric graph neural networks (GNNs) as general molecular/amino acid descriptors containing all-atom information.
>
> Our paper addresses a previously unanswered research question in graph representational learning:
> **"Are pre-trained all-atom geometric GNN representations transferable to protein modeling, and how expressive are they?"**
>
> Rather than extending denoising pre-training techniques or focusing on pre-training task choices or model architectures, our contributions are as follows:
>
> 1. **Scaling Study of Geometric GNNs:**
>    We examined the scaling behaviors of state-of-the-art geometric GNNs across unsupervised, self-supervised, and supervised setups, focusing on their general applicability rather than developing new architectures or pre-training objectives.
>
> 2. **Transferability to Protein Modeling:**
>    We pre-trained these GNNs on small molecular datasets and demonstrated their transferability to protein modeling tasks with all-atom resolution, showcasing their expressiveness in these settings.
>
> Per the reviewers’ suggestions, we restructured the abstract and introduction to emphasize this research question.
>
> To explore the expressiveness of these representations, we thoroughly evaluated their performance and scaling across various tasks, including:
> - **Kinetic Modeling/Markov State Modeling**
> - **Protein Folding Classification**
>   (trained on equilibrium small molecules, transferred to non-equilibrium and equilibrium protein structures)
> - **Molecular Force Field and Property Predictions**
>   (both fine-tuning and training from scratch)
>
> ## Response to Concerns About Limited Molecular Diversity in Evaluation
>
> We thank the reviewer for highlighting this important concern. We agree that QM9, while historically significant, is a limited benchmark due to its age and the prevalent use of random splitting strategies, which favor memorization over generalization. To address this, we adopted scaffold splitting (Page 8, Table 2), as MoleculeNet [1], to better evaluate generalization across molecular scaffolds. Under this more rigorous evaluation, our results show that increasing the number of parameters does not consistently improve performance, in contrast to prior findings.
>
> We also concur that **xxMD**, which includes non-equilibrium molecular trajectories, is a more informative benchmark than MD17/rMD17. As noted in the xxMD paper [2] (Figure 2 and S4), random splits of poorly sampled trajectories result in minimal distinction between training and test sets.
>
> While we appreciate the reviewer’s suggestions to explore other benchmarks such as Polaris and TDC, these datasets provide only 2D SMILES strings, which are incompatible with geometric GNNs that require 3D molecular information. Furthermore, we are unaware of prior work benchmarking geometric GNNs on these datasets. Single conformers in drug-like property prediction pose additional challenges due to stereochemistry (e.g., chirality) significantly influencing functionality. The limited availability of 3D molecular benchmarks reflects the current focus of all-atom geometric GNNs on quantum chemical property prediction.
>
> To address these gaps, we provide:
> - Insights on pre-training GNNs and leveraging their embeddings for downstream tasks.
> - Guidance on integrating these embeddings with other architectures in protein modeling (Figure 2/3/14–20, Table 1).
> - A new section (Appendix M, Page 31–32) discussing optimal model configurations and dataset splitting strategies for robust evaluation.
>
> [1] Wu, Zhenqin, Ramsundar, Bharath, Feinberg, Evan N., Gomes, Joseph, Geniesse, Caleb, Pappu, Aneesh S., Leswing, Karl, and Pande, Vijay. "MoleculeNet: a benchmark for molecular machine learning." Chemical Science, vol. 9, no. 2, 2018, pp. 513–530. Royal Society of Chemistry.
> [2] Pengmei, Zihan, Liu, Junyu, and Shu, Yinan. "Beyond MD17: the reactive xxMD dataset." Scientific Data, vol. 11, no. 1, 2024, p. 222. Nature Publishing Group UK London.

---

> ### Author Response · Authors · 2024-11-20
> **Response to Reviewer GND7 Part 2**
>
> ## Response to Concern About Lack of General Evaluations
>
> We thank the reviewer for raising this point. Our study thoroughly examines the performance and scaling behavior of pre-trained GNNs across diverse tasks, including:
>
> - **Single-molecule conformational variety:** Force field regression and kinetic modeling.
> - **Multi-molecule chemical variety:** Quantum property prediction.
> - **Peptide/protein conformational variety:** Kinetic modeling.
> - **Protein biological variety:** Folding classification.
>
> We present over 100 pre-training experiments, testing two GNN architectures with varied depth, width, aspect ratio, and radius cutoff configurations on equilibrium and non-equilibrium datasets. Key experiments include:
>
> 1. **VAMPNet:** Three systems (molecules to proteins) tested with different embedding dimensions and token mixers.
> 2. **Folding classification:** ProNet (with/without pre-trained all-atom embeddings) tested across dimensions, representing the most comprehensive study in relevant literature.
> 3. **Force field prediction:** Two GNNs (non-pre-trained and pre-trained on PCQM and Denali) evaluated across hidden dimensions on 4 molecular systems from xxMD.
> 4. **Molecular property prediction:** Two GNNs (non-pre-trained and pre-trained on PCQM) evaluated across hidden dimensions on 5 tasks.
>
> ### Key Insights About Scaling
>
> #### **Embedding Transfer for Protein Modeling**
> Pre-trained embeddings without token mixing (e.g., simple summation) show fictitious scaling benefits due to higher-dimensional embeddings retaining more information. Introducing token mixers (e.g., MLPs or Transformers) enables structural unit interdependence modeling, significantly boosting performance. While larger models show slight gains with mixers, there are no sustained scaling benefits, as confirmed by folding classification results (Page 30, Figure 20).
>
> #### **Label Uncertainty Bottlenecks**
>
> 1. **xxMD:** Pre-training benefits vary. ViSNet, already near the label uncertainty limit, shows minimal pre-training gains (2 of 4 sets benefit), while the ET architecture shows consistent gains across all 4 sets (Page 9, Table 3; Page 32, Table 9). Scaling data, rather than models, proves more impactful in low-data regimes.
>
> 2. **QM9:** For tasks like quantum chemical property prediction (e.g., homo-lumo gap), the inherent uncertainty in labels limits test-set accuracy regardless of pre-training or model expressiveness. For example, common density functionals exhibit ev-scale differences (Page 16, Figure 5), making milli-ev-scale accuracy unrealistic for QM9.
>
> ### Additional Data Provided
>
> We also include xxMD-temporal benchmark data (dimensions 64–384) across all setups (non-pre-trained, pre-trained on PCQM/Denali) in the appendix (Page 32, Table 9).

---

> ### Author Response · Authors · 2024-11-23
> **Further discussion**
>
> Dear Reviewer GND7,
>
> We just would like to ask if our response has addressed your concerns? If not, we are very happy to discuss and further clarify.
>
> Kind Regards,
> Authors

---

> > ### Comment · Reviewer_GND7 · 2024-11-24
> >
> > I thank the authors for their careful response, including restructuring and clarifying their research question, which addresses my primary feedback. I am happy to increase my score.
> >
> > A small point of clarification, TDC (https://tdcommons.ai/overview) is a collection of benchmarks beyond small molecules, and includes tasks for macromolecules and peptides. Additional clarifications around the limitations and scope of QM9 address my concerns.

---

> > > ### Author Response · Authors · 2024-11-24
> > > **Thank you**
> > >
> > > We really appreciate the reviewer for helping us improving the manuscript quality. Wish you a good break :)
> > >
> > > Kind Regards,
> > > Authors

---

### Author Response · Authors · 2024-11-20
**Global Rebuttal**

We sincerely thank the reviewers for their thoughtful and constructive feedback, which has greatly contributed to improving the quality and clarity of our manuscript. Below, we summarize the major updates and changes made in response to the reviewers’ comments:

### Summary of Changes and Key Clarifications:

1. **Research Question and Contributions:**
   - We emphasized our central research question:
     **"Are pre-trained all-atom geometric graph neural network (GNN) representations transferable to protein modeling, and how expressive are they?"**
   - To address this, we studied the scaling behaviors of state-of-the-art geometric GNNs in unsupervised, self-supervised, and supervised setups and demonstrated their transferability to proteins with all-atom resolution.

2. **Figure Revisions:**
   - Based on the feedback from **Reviewer i3nd**, we revised Figures 4, 6, and 7 for improved clarity by annotating legends with specific model depths and widths where relevant. These updates are reflected in the manuscript.

3. **Restructuring the Introduction and Abstract:**
   - Per suggestions from multiple reviewers, we restructured the introduction and abstract sections to better highlight our research question, contributions, and the relevance of this study.

4. **Additional Experimental Results:**
   - Responding to feedback from **Reviewer i3nd**, we added experimental results comparing the effect of model depth and aspect ratio during fine-tuning on QM9. The results, summarized in **Appendix L, Table 10 (Page 32)**, show a general trend where pre-trained deeper models perform better than shallower ones, aligning with findings in prior literature (e.g., Li et al. 2024).

5. **Clarifications on Zero-Shot Transfer and Pre-Training Objectives:**
   - We addressed concerns from **Reviewer KCDU** and **Reviewer 5LXW** about the definition of zero-shot transfer learning. Our setup involves pre-training on small molecules and directly transferring embeddings to infer atomistic representations for proteins, which is then combined with separate tasks. We clarified why this qualifies as zero-shot embedding inference, distinct from fine-tuning.
   - We elaborated on the rationale behind selecting coordinate denoising as the pre-training objective for its simplicity and effectiveness.

6. **Benchmarks and General Evaluations:**
   - We addressed concerns about dataset diversity by discussing scaffold splitting in QM9 and incorporating non-equilibrium datasets like xxMD. These ensure rigorous evaluation of generalization capabilities.
   - Additional data on xxMD datasets comparing pre-trained and non-pre-trained models were provided in **Appendix L, Table 9 (Page 31)** to better support claims about model performance across dimensions and molecular systems.

### Final Remarks

We hope these revisions and clarifications address the reviewers' concerns and further demonstrate the novelty and importance of our work. We remain grateful for the reviewers' insightful feedback, which has strengthened our manuscript significantly.

Thank you for your time and consideration.

---

### Comment · Area_Chair_8tuF · 2024-11-22

Hi reviewers,

The authors have posted their rebuttals. Could you please check their responses and engage in the discussions? Please also indicate if/how their responses change your opinions.

Thanks,

AC

---

### Meta-Review · Area_Chair_8tuF · 2024-12-20

**Metareview:**

This paper studies if the pre-trained all-atom geometric GNN representations are transferable to protein modeling and their expressivity.
To answer this, the paper studies the scaling behaviors of state-of-the-art geometric GNNs in unsupervised, self-supervised, and supervised setups and demonstrates their transferability to proteins with all-atom resolution. The authors explore many aspects of geometric GNNs, like model size, aspect ratio, nearest neighbor cutoff radius, architecture, and transferability among different data types, which is comprehensive and distinguishes the work from other studies. Overall, I think the work is interesting, novel, and useful to the field of geometric GNNs. Thus, an acceptance is recommended.

**Additional Comments On Reviewer Discussion:**

Among four reviewers, three acknowledged their concerns were successfully addressed during the discussion period. The other reviewer 5LXW had a long discussion with the authors. In her/his last comment, Reviewer 5LXW acknowledged she/he is not an expert in the field, so she/he did not want to change initial ratings ( except the overall rating which is 6) but didn't wish to block the potentially useful publication as well. I checked all the discussions and believe most of the concerns have been addressed.

---

### Decision · Program_Chairs · 2025-01-22

Accept (Poster)